# Confinement in crystal lattice alters entire photocycle pathway of the Photoactive Yellow Protein

Patrick E. Konold[1,3], Enis Arik[1,3], Jörn Weißenborn[1], Jos C. Arents[2], Klaas J. Hellingwerf [2], Ivo H. M. van Stokkum [1], John T. M. Kennis [1] & Marie Louise Groot [1✉]

Femtosecond time-resolved crystallography (TRC) on proteins enables resolving the spatial structure of short-lived photocycle intermediates. An open question is whether confinement and lower hydration of the proteins in the crystalline state affect the light-induced structural transformations. Here, we measured the full photocycle dynamics of a signal transduction protein often used as model system in TRC, Photoactive Yellow Protein (PYP), in the crystalline state and compared those to the dynamics in solution, utilizing electronic and vibrational transient absorption measurements from 100 fs over 12 decades in time. We find that the photocycle kinetics and structural dynamics of PYP in the crystalline form deviate from those in solution from the very first steps following photon absorption. This illustrates that ultrafast TRC results cannot be uncritically extrapolated to in vivo function, and that comparative spectroscopic experiments on proteins in crystalline and solution states can help identify structural intermediates under native conditions.

[1] Department of Physics and Astronomy and LaserLaB, Faculty of Science, Vrije Universiteit, De Boelelaan 1081, 1081 HV, Amsterdam, The Netherlands. [2] Laboratory for Microbiology, Swammerdam Institute for Life Sciences, University of Amsterdam, Science Park, 1098 XH Amsterdam, The Netherlands. [3]These authors contributed equally: Patrick E. Konold and Enis Arik. ✉email: m.l.groot@vu.nl

Time-resolved macromolecular crystallography (TRC) enables direct visualization of a protein structure while exerting its function, e.g., during signaling, ion translocation and catalysis[1–7], and with the advent of free-electron lasers, timescales of femto- to picoseconds are now experimentally accessible[4,8–15]. Consequently, protein structural dynamics can now be studied with both TRC and various ultrafast transient spectroscopic methods. Vibrational spectroscopy offers, after successful assignment, high resolution structural information on interatomic distances and dihedral angles of catalytically active groups[16]. Therefore, the combination of TRC and spectroscopic techniques enables biophysical interrogation with the utmost resolution. A looming pitfall in this pursuit concerns the sample treatment in each method. Spectroscopy is routinely performed on proteins dissolved in aqueous buffer solution of known pH—largely consistent with their native state in vivo. For TRC in contrast, proteins are tightly packed in a regular lattice with lower relative hydration compared to the solvated state in vivo. Hence, the question arises how these conditions influence the (photo)chemical reactions and how the ensuing structural transformations vary within the two environments.

Here, we investigated the full photocycle dynamics of a small signal-transduction protein, Photoactive Yellow Protein (PYP), responsible for the negative phototactic response in the photo-trophic bacterium *Halorhodospira halophila*[17]. This protein is an excellent system to study the effect of crystallization on experimental photo-transformations, as PYP is relatively small (14 kDa), biochemically stable, water-soluble, and is especially relevant because it has served as a model system for the earliest TRC studies[1–3,5,11,15]. We report electronic and vibrational transient absorption measurements spanning the full PYP photocycle in crystalline form and dissolved in buffer solution.

## Results

**Initiating the PYP photocycle**. PYP has an intrinsic chromophore, *para*-hydroxycinnamic acid (pCa)—an extensively conjugated anion, which is covalently attached to a cysteine residue (Cys69) through a thioester linkage to the protein backbone[18]. Upon blue light absorption ($\lambda_{max} = 446$ nm), PYP enters a photocycle involving several intermediates with lifetimes ranging from femtoseconds to seconds, culminating in a large secondary structural change that underlies its signaling function. We initiated the photocycle by excitation of the pCa with a 50 fs laser pulse centered at 475 nm and recorded electronic transient absorption measurements from 100 fs to 0.3 ms using electronically-synchronized femtosecond lasers and augmented these data, with data from Yeremenko et al.[19], collected from microseconds to 1 s, to span the full photocycle dynamics. We used a low excitation power of 400 nJ, focused to a spot size of 250 µm in diameter, which, centered on the flank of the absorption band of PYP, resulted in 5% excited proteins, thus avoiding multi-photon and/or photo-ionization processes[20], for both PYP in the crystalline form ($PYP_C$) and in solution ($PYP_S$).

**Transients in the UV-visible spectral region**. The electronic responses of $PYP_C$ and $PYP_S$ were recorded in the 380–570 nm spectral window, and primarily reveal photochemical changes to the pCa chromophore. Figure 1a, b shows a selection of the collected absorption difference spectra for $PYP_S$ and $PYP_C$ and panel C provides a more detailed look at the kinetics at three selected wavelengths, comparing the dynamics of $PYP_S$ (black) and $PYP_C$ (red). Additional time traces are shown in Supplementary Figs. 1 and 2. The time traces reported in Yeremenko et al.[19] have been overlaid with those recorded here, which

resulted in an excellent agreement between the two separate experiments in the overlap region between 10–300 µs.

Upon excitation, an immediate buildup of a mixed signal composed of excited state absorption ($\lambda^{max} \approx 400$ nm), bleached ground state absorption ($\approx 450$ nm) and stimulated emission ($\approx 500$ nm) is observed. With increasing pump-probe delay time, the excited state and stimulated emission signals decay and product formation is observed in the 480–500 nm region. Then, on a sub-millisecond timescale, the product signal at $\approx 480–500$ nm decays, the ground state bleach shows an increase in signal, due to disappearance of the compensating red-product absorption, and a small product band appears <400 nm (not fully covered in Fig. 1a, b). Notable differences between the traces of $PYP_S$ and $PYP_C$ exist on all timescales. For example, the dynamics on the (sub-)picosecond timescale appear different, with a faster initial decay in $PYP_C$, while the yield of photoproduct formation in $PYP_C$ is lower than in $PYP_S$. Differing dynamics were also observed on the 10–20 ns timescale that is absent in $PYP_S$. Spectral differences between the solution and crystalline samples are also present, these are most likely due to the higher absorption in the crystals, diminishing the bleach signals at 440 nm and to increased scatter, which leads to a certain degree of distortion of the absorption difference spectra. The steady state absorption spectra of $PYP_S$ and $PYP_C$ differ only slightly, crystalline red-shifted (3–4 nm) with a similar width and overall shape (Supplementary Fig. 3), in agreement with earlier observations[19,21,22].

**Transients in the mid-IR spectral region**. We further recorded the absorption changes induced in the mid-IR region between 1780 and 1530 cm$^{-1}$ upon excitation at 475 nm for $PYP_S$ and $PYP_C$ (Fig. 2). The vibrational response in the mid-IR region enables precise assignment of structural dynamics that occur throughout the photocycle; specifically, changes to the hydrogen bond network surrounding the chromophore, changes in the chromophore structure and concerted changes of the protein backbone.

Time traces collected in the mid-IR region up to 0.3 ms show a similar divergence between the two PYP forms as in the visible spectral region. Figure 3 presents time traces at selected frequencies: 1668 cm$^{-1}$, the dynamics forming a positive feature are delayed in the crystal with respect to solution, and at 1686 cm$^{-1}$ the rise is more complex in the crystal than in solution. As will be argued in more detail below based on $^{13}C$ isotope labeled measurements, these frequencies track the pCa carbonyl, and record the breaking of the hydrogen bond between the pCa carbonyl and Cys69, with its precise position being dependent on the environment of the carbonyl. See Supplementary Figs. 4–6 for a complete representation of the collected time traces.

**Photocycle analysis**. To resolve the individual steps and intermediates in the PYP photocycle, all time traces were fitted to a target model[23–25], analogous to models previously used for analysis of TRC dynamics[2,3,11]. Strikingly, we found that the visible and mid-IR data of $PYP_S$ in solution could be described with the same model (Fig. 4), with excellent fit quality. This indicates that the electronic and vibrational absorption changes display identical dynamics, which enables direct comparison of the species associated difference spectra (SADS) obtained with these two types of spectroscopy. Likewise, the visible and mid-IR $PYP_C$ data could be fitted with the same kinetic model. However, as compared to the $PYP_S$ data, according to the divergent kinetics discussed above, different parameters were required to achieve an adequate fit for each $PYP_S$ and $PYP_C$, and, notably, the crystalline

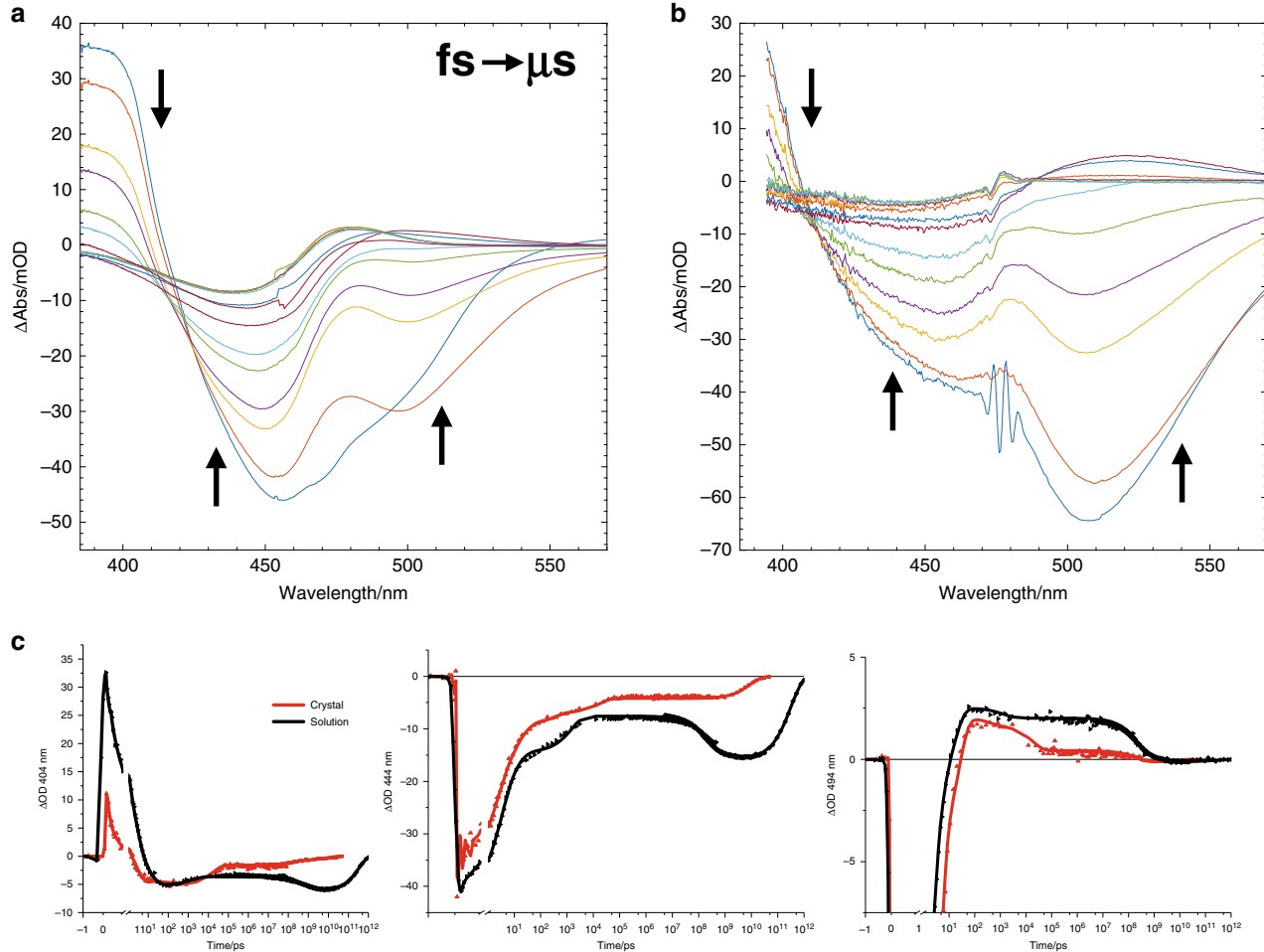

**Fig. 1 Transient absorption data in the visible collected over 12 decades in time.** A selection of transient absorption spectra as a function of time delay are shown, upon excitation with a 50-fs laser pulse centered at 475 nm, for **a** PYP$_S$ and **b** PYP$_C$. The temporal evolution is indicated by the black arrows. In PYP$_C$, a scattered-light artifact is visible at the wavelength of excitation. **c** Wavelength traces at 404, 444 and 494 nm of PYP$_S$ (black) and PYP$_C$ (red), see Supplementary Figs. 1 and 2 for traces at additional wavelengths. The data from the current study with sub-picosecond time resolution has been merged with data taken from Yeremenko et al.[19] with microsecond time resolution, with the latest time point at 48 ms (crystal) or 1 s (solution). Wavelength is indicated in the ordinate label. Note that the time axis is linear until 1 ps (after the maximum of the instrument response function (IRF)), and logarithmic thereafter. A single scaling parameter has been used to connect the data from the experiments with sub-picosecond and microsecond time resolution. PYP$_S$ was measured at pH 8, PYP$_C$ at pH 6.5. PYP$_S$ time traces collected at pH 6 are shown in Supplementary Fig. 1 and exhibit corresponding dynamics.

kinetic model included an additional intermediate state on the nanosecond time scale (Fig. 4).

Figure 4 shows the target models used for the solution and crystallized states. The nomenclature for PYP photocycle intermediates varies throughout the literature, and since distinct dynamics are observed here for PYP$_C$, we use the symbol "I$_X$" for notation of PYP$_S$ intermediate states, and "pC" (with C for color, i.e., red or blue) for those in PYP$_C$, except for the final pB states. Previous pump-dump-probe, transient absorption and fluorescence studies[24,26–34] were used to define and confine our target model. With target analysis, one can extract the characteristic spectra of the photocycle states, namely, contributions from the excited state (ES), primary photoproduct (I$_0$), and subsequent intermediates. The target model in Fig. 4 describes the experimentally observed key elements of the PYP photocycle[23,24,28,31,33–35] such as the multi-phasic decay of the excited-state and the involvement of a ground state intermediate, and allows for estimation of the quantum yield of isomerization, which has earlier been found to be highest from the shorter-lived excited states[24]. The multi-phasic decay of the excited state is fitted by a tri-exponential decay from three excited states ES$_1$, ES$_2$, ES$_3$ populated sequentially, with a single ES

spectrum[23,24]. Both a sequential model, in which each short-lived excited state relaxes into the next longer-living state, and a parallel model, in which the excited states decay independently, give an equally good fit to the data. It is therefore up to now unclear whether the different time constants and their different reactivity are representative of a relaxation process in the excited state (i.e., an evolution from a more- towards a less-productive minimum on the excited state surface), or whether they are due to heterogeneity or inhomogeneity, due to small structural differences between proteins[36,37].

Each ES can decay into the ground-state intermediate (GSI), and via that return to the ground state, and into the isomerized pR$_0$/I$_0$ state(s). GSI is a short-lived component, which decays into the stable ground state on a picosecond time scale without giving rise to formation of (a) transient photocycle intermediate(s)[23,35]. The UV–Vis species associated difference spectra (SADS) that result from this analysis are shown in Fig. 5. The I$_0$, I$_1$, pR$_0$, pR$_1$, pR$_2$ intermediates display red shifted product absorption, typical for the pCa *cis*-isomer, whereas the pB$_1$, pB$_2$, pB$_{crystal}$ intermediates display blue-shifted product absorption, typical for the protonated pCa (*cis*-isomer).

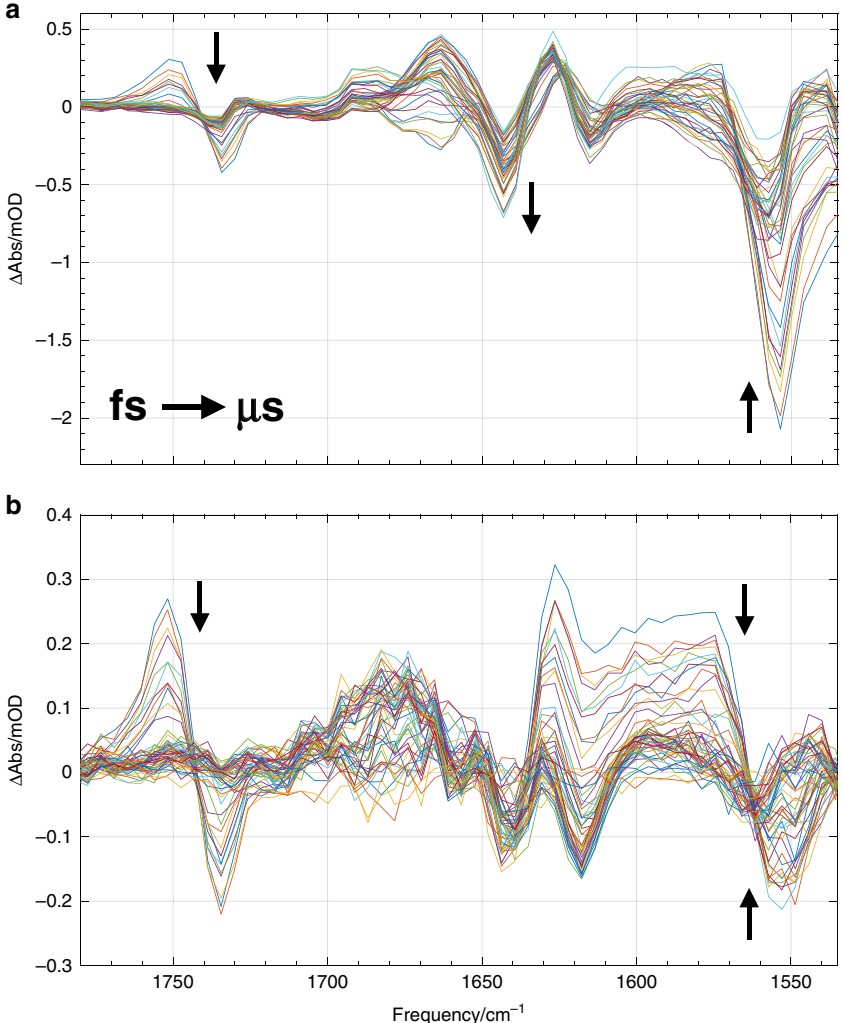

**Fig. 2 Transient absorption difference spectra in the midinfrared.** A selection of transient absorption difference spectra in the mid-IR spectral region following excitation with 475 nm light for **a** PYP_S and **b** PYP_C, from 100 fs to 100 ms. Full spectra were recorded for each laser pulse after dispersing the probe light on a 64 element Mercury Cadmium Telluride photodiode array, resulting in a ~4 cm$^{-1}$ sampling resolution. Approximate temporal evolution from femtoseconds to microseconds is denoted by the black arrows.

In accordance with the observed differences in the time traces, fitting of the data against the target model yields different rates for $PYP_S$ and $PYP_C$, and an extra intermediate was required for $PYP_C$. The initial decay of the ES in $PYP_C$ is fastest, 0.35 ps, but results in only 10% $pR_0$. A second decay phase in the ES leads to further $pR_0$ formation in 2.6 ps, resulting in an overall quantum yield of 23% (Fig. 4). For clarity, we also present a simplified version of the photocycle in Fig. 6. There, we indicate the excited state process of $PYP_C$ with a weighted average of 1 ps. In $PYP_S$, 29% of $I_0$ is formed in 0.6 ps and another 2% in 2 ps. We summarize this as, formation of the first *cis* isomer of pCa, with red-shifted product absorption, is slower in $PYP_C$ than in $PYP_S$ (1 vs 0.6 ps) and is formed with a lower quantum yield (0.23 vs 0.31). In the formation and dynamics of the initial red-shifted *cis*-isomer photoproducts in $PYP_C$ an additional intermediate is present: $pR_0$, $pR_1$, $pR_2$ vs $I_0$ and $I_1$ in $PYP_S$. $pR_0$ has a shorter lifetime than $I_0$ (0.3 vs 1 ns) and $pR_1$ decays into $pR_2$ in 18 ns. The formation of pB-like states occurs on similar timescales in $PYP_S$ and $PYP_C$, but with a lower quantum yield, as only half of the $pR_2$ states form $pB_C$, resulting in a pB yield of 0.3 vs 0.11 in $PYP_S$ and $PYP_C$ respectively. Recovery of the ground state takes 9 ms in $PYP_C$ and occurs with biphasic time constants of 1.3 and 320 ms in $PYP_S$[19].

The $PYP_S$ photocycle kinetics are in agreement with earlier spectroscopic reports[19,23–25,27,28,35,38–45], while the $PYP_C$ results closely follow those observed in TRC by Schotte[2], Jung[3] and Ten Boer et al.[11] (Note that these latter three studies mutually agree on the data, but differ in the bond-order of the double bond that can isomerize in the chromophore, due the use of DFT optimized structure vs non-DFT optimized structures[46]. In a recent ab initio computational study, the former structure was favored[47]). The agreement between the current data and those of TRC includes the dynamics on the 18 ns timescale associated with formation of $pR_2$, 100% yield of $pR_2$ from $pR_0$[2] and the 22% overall pR yield[11], though the former reported the $pR_2$ to pB transition to take place in 410 ms. The results for $PYP_S$ and $PYP_C$ clearly demonstrate an effect of the crystallization on the PYP photocycle, including on the earliest photocycle events.

**Assignment using site-specific isotope labeling in mid-IR**. The SADS in the mid-IR spectral range of $PYP_S$ and $PYP_C$ (Fig. 7) allow for a more detailed structural analysis of the photocycle intermediates. Of particular interest is the timescale of the breaking of the hydrogen bond between pCa carbonyl and the Cys69 residue and its role in facilitating the isomerization process[2,3,11,35]. However, the frequency of pCa carbonyl in the $I_0$

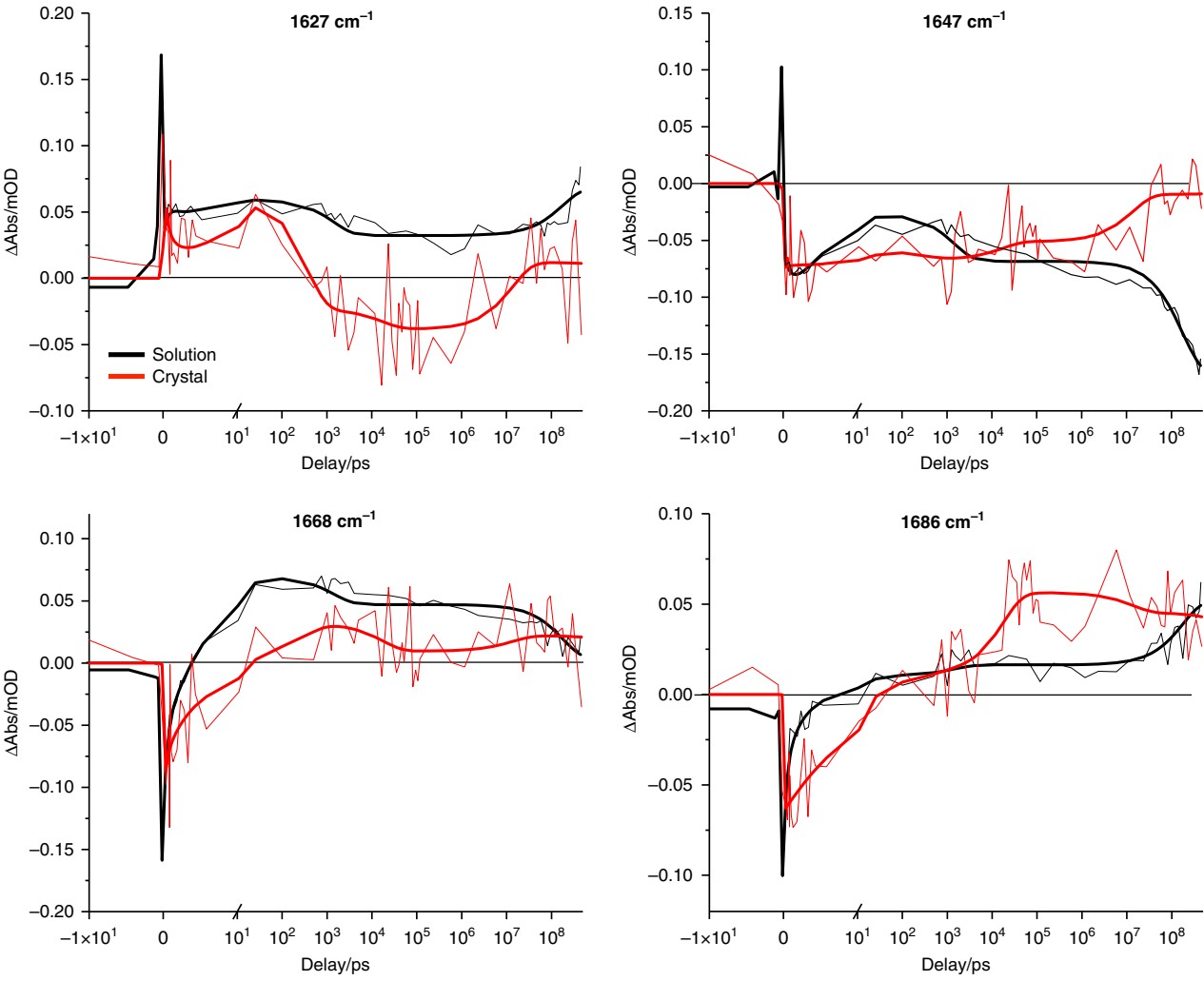

**Fig. 3 Transient absorption time traces in the midinfrared.** Time traces at selected wavenumbers of $PYP_S$ (black) and $PYP_C$ (red). Note that the time axis is linear until 10 ps (after the maximum of the IRF), and logarithmic thereafter. The $PYP_S$ data was scaled lower by a factor of 5 to facilitate comparison.

state has been a matter of contention[35,48]. To improve the mid-IR spectral assignment, we utilized a $^{13}C$ isotopic label at position C9 of the pCa chromophore ($^{13}C=O$ pCa– PYP, see atom numbering in Fig. 6b). In doing so, the pCa carbonyl mode is expected to downshift, in addition to all modes involving C9, following earlier Raman analysis[49,50]. Unlabeled PYP and $^{13}C=O$ – labeled PYP samples in solution were measured consecutively in a single experiment and repeated twice to ensure reproducibility. Both datasets were modeled according to the schemes described above, with all time constants fixed. In Fig. 8, we report the mid-IR SADS of both samples, where positive bands originate from product-state absorption due to the formation of the PYP photoproducts: ES, $I_0$, $I_1$, pB, and negative bands are due to the disappearance of ground state bands. The unlabeled SADS are consistent with those previously reported[35,38,39]. In Supplementary Table 1, we summarize the assignments of the spectral features observed in Figs. 7 and 8 that can be made based on vibrational spectroscopies and DFT calculations of the pCa chromophore, various isotope-labeled and point mutants of PYP previously reported[35,39,44,45,49–54].

Comparison of the unlabeled and labeled $^{13}C=O$ $PYP_S$ spectra reveals a large isotope effect for the 1664/1643 $cm^{-1}$ positive/negative bands appearing in the $I_0$ and $I_1$ SADS and at 1688/1643 $cm^{-1}$ in $pB_1$ (Fig. 8). This confirms that this feature is due to the pCa carbonyl, with the upshift reflecting breaking of

the hydrogen bond between the pCa carbonyl and the protein upon isomerization. Note that part of the 1664/1643 $cm^{-1}$ band shows no isotope effect and must therefore arise from Amide I C = O oscillators of the protein backbone. In addition, a large decrease in the amplitude of the 1554 $cm^{-1}$ band is observed for the ES, $I_0$ (1558 $cm^{-1}$), $I_1$ states. This is in agreement with the assignment to the $C7=C8$ mode (Supplementary Table 1), and the observation of this band to be reduced in amplitude and downshifted by ~5–10 $cm^{-1}$ in $^{13}C=O$—PYP ground-state Raman spectra[50,55].

**Structural changes occurring during the photocycle of $PYP_S$ and $PYP_C$.** The mid-IR SADS of $PYP_C$ in Fig. 7 are qualitatively similar to those of $PYP_S$, however, they exhibit several differences in quantitative details: in the pCa carbonyl region, a negative double band structure at 1655 and 1635 $cm^{-1}$ in $pR_1$ and $pR_2$ is indicative of heterogeneity in the pCa–Cys69 hydrogen bond strength in the ground state. In the $pR_0$ state, the 1660–1700 $cm^{-1}$ region shows no positive band, suggesting that the hydrogen bond of the pCa carbonyl with Cys69 is intact in this intermediate. This is consistent with TRC structures reported for $pR_0$ where pCa is highly contorted with its carbonyl oriented ≈90° out of plane with the phenolate, and the hydrogen bond intact[2,3]. In $pR_1$, a minor positive band at 1670 $cm^{-1}$ has appeared, signaling

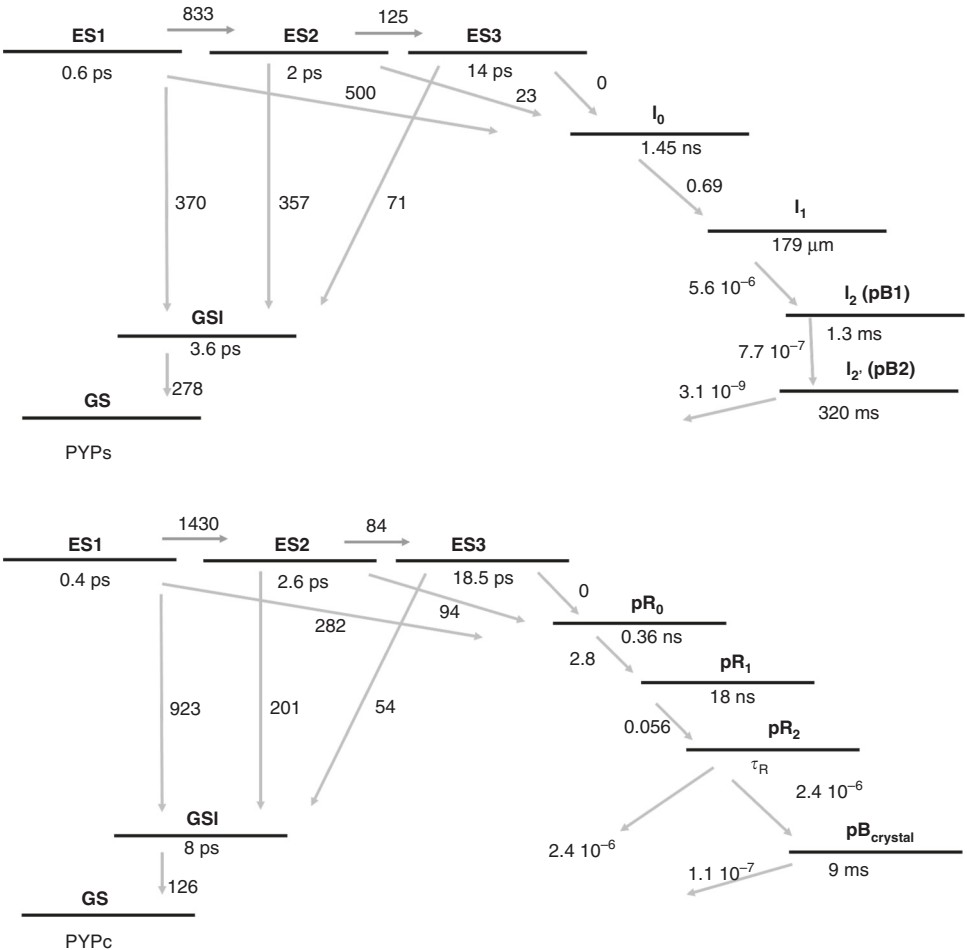

**Fig. 4 Kinetic schemes of the photocycle of PYP in solution and in crystalline form.** Top panel: PYP in solution, bottom panel: PYP in crystalline form. Upon absorption of light by pCa in the ground state (GS), the excited state is formed from which the system transitions through the reaction. The relevant states are indicated by the following abbreviations: $ES_{1-3}$: excited states sharing the same spectrum, evolving sequentially from 1→2→3; GSI: ground state intermediate; $I_0$, $I_1$, $I_2$ (pB1), $I_{2'}$(pB2), $pR_0$, $pR_1$, $pR_2$, $pB_{crystal}$: photocycle intermediates. Note that we use the symbol "$I_X$" for notation of $PYP_S$ intermediate states, and "pC" (with C for color, i.e., red or blue) for those in $PYP_C$, except for the final pB states. The transition rates indicated near the arrows are in $ns^{-1}$, the lifetime of each state is indicated in italic. The rate constants involving the ES and GSI for $PYP_S$ have been fixed from ref. [24], all other parameters have been estimated. Note that in the crystal as evidenced from the UV–Vis data[19] half of the $pR_2$ states is converted to pB, whereas the other half branches directly to the ground state, with a rate of $k_{R,VIS}$ $2.4 \times 10^{-6}$. Note that the longer time scales (after 100 ms) are based solely on the the UV–Vis data of ref. [19]. Concentration profiles computed with these kinetic schemes for the photocycle intermediates states are shown in Supplementary Fig. 7.

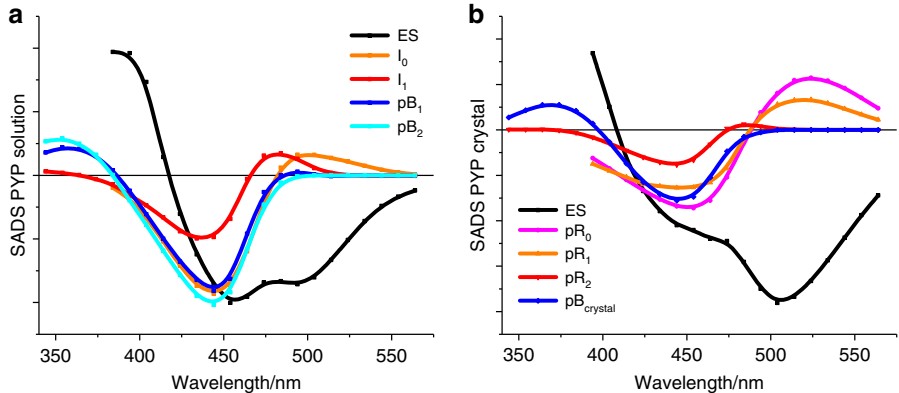

**Fig. 5 UV–Vis SADS of PYP.** The SADS in solution (**a**) and crystalline (**b**) form have been resolved from fitting the target model shown in Fig. 4 to the experimental data, and display the bleached ground state, and stimulated emission, as negative features and induced absorption as positive features. The characteristic blue (≈370 nm) absorption of pCA in the pB states signals protonation of pCa. The region of the ground state bleach in the $PYP_C$ dataset appears to be diminished with respect to that in $PYP_S$; this is an effect of the high absorption of the crystals around 444 nm. Note that data from Yeremenko et al.[19] has been added to the analysis to more reliably estimate the dynamics and SADS on the ~10–300 ms timescale and to span the full photocycle dynamics up to 1 s. Source data are provided as a Source Data file.

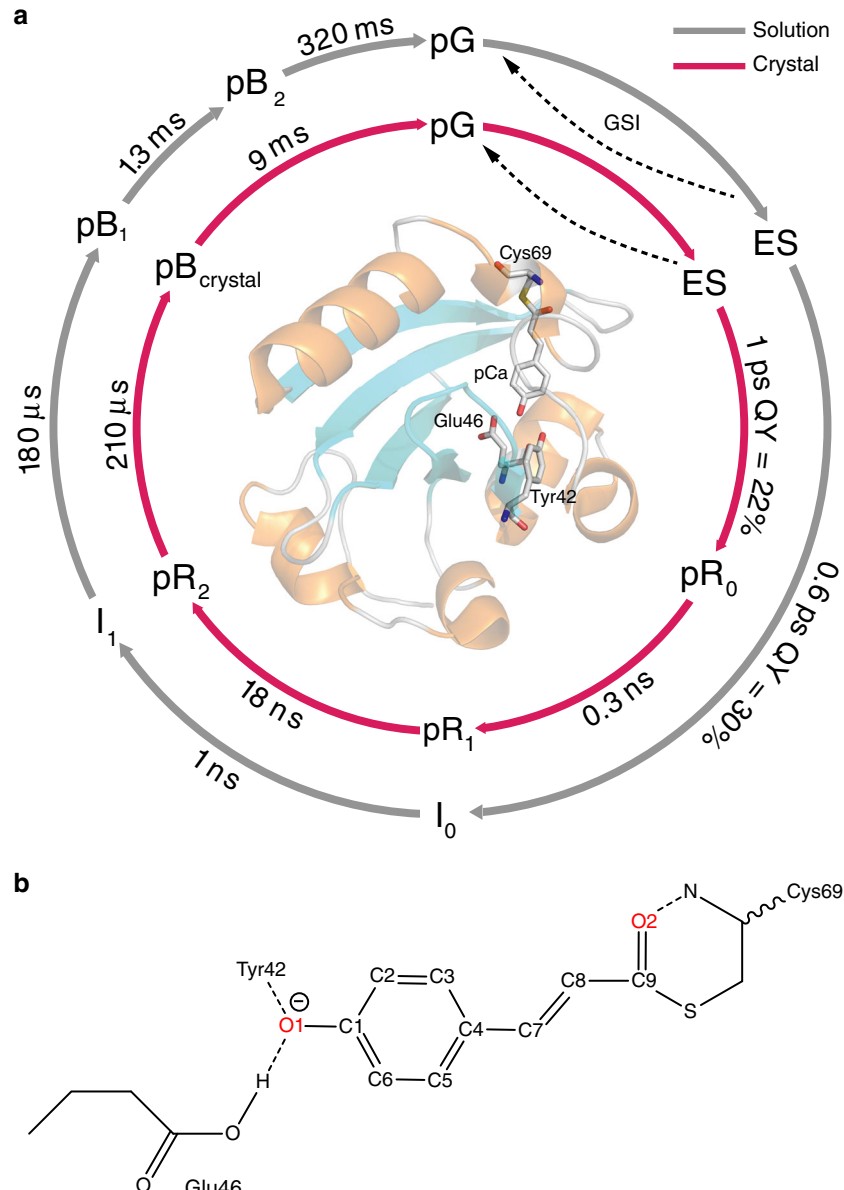

**Fig. 6 Photocycle scheme of PYP in solution and in crystalline form and pCa structure. a** Simplified photocycle scheme for PYP$_C$ (red arrows) and for PYP$_S$ (gray arrows). The full kinetic schemes for PYP$_S$ and PYP$_C$ are shown in Fig. 4. Upon absorption of light by pG, the excited state is formed (ES), from which pCa trans-to-cis isomerization takes place in 0.6 ps in PYP$_S$ and ~1 ps in PYP$_C$, with quantum yields of 30 and 23% respectively. In PYP$_C$, an additional intermediate is formed, pR$_0$, followed by pR$_{1,2}$ respectively, in comparison to the intermediates (I$_{0,1}$) in PYP$_S$. **b** pCa molecular structure in trans-configuration and interactions with neighboring sidechains. Hydrogen bonds are indicated by dashed lines.

breaking of the hydrogen bond on the 0.3 ns timescale in at least a subset of the protein molecules (Figs. 2 and 7). The pR$_2$ spectrum, formed in 18 ns, shows a band at 1688 cm$^{-1}$, very similar to that of pB of PYP$_S$. Note, however, that in PYP$_S$ this band is formed only after 180 ms in pB (see corresponding time trace in Fig. 3). A shift of the free pCa carbonyl band from 1664 cm$^{-1}$ in I$_1$ to 1688 cm$^{-1}$ in pB, assigned with the help of the isotopic labeling (Fig. 8), we suggest is due to the pCa carbonyl residing in a different environment, possibly more hydrophobic in pB compared to I$_1$. Note that calculations on vibrational frequencies of PYP[49] predict the pCa carbonyl to be at the relatively high frequency of 1688 cm$^{-1}$ for the blue-shifted protonated product.

Changes in the region of the protonated carboxyl of Glu46 (1740–1760 cm$^{-1}$) in PYP$_C$ indicate that the hydrogen bond between the pCa phenol ring and Glu46 is relaxed throughout the photocycle of PYP$_C$, in agreement with the reported lengthening

from 2.50 to 2.94 Å in the TRC study of Pande et al[5], but in contrast to the observed strengthening in the solution PYP$_S$ I$_0$ and I$_1$ states[35,38]. The absorption changes in the Glu region disappear in pR$_2$ and pB in PYP$_C$, suggesting that the ground state configuration of Glu46 is recovered and that it remains protonated, even though the observed blue-shift of the chromophore electronic absorption in the PYP$_C$ pB state signals chromophore protonation. This observation suggests that the proton donor to pCa in PYP$_C$ is not Glu46. Indeed, Schotte et al.[2] reported that in pB, Arg52 switches to an "open" conformation exposing the pCa phenolate and facilitating its protonation with a water molecule hydrogen-bonded to the phenolate and the protein backbone. Note that in the mid-IR study the timescale extended out to 0.3 ms; therefore, the protein structural changes associated with formation of the signaling state in PYP$_S$ are not included. Hence, some further spectral differences associated with

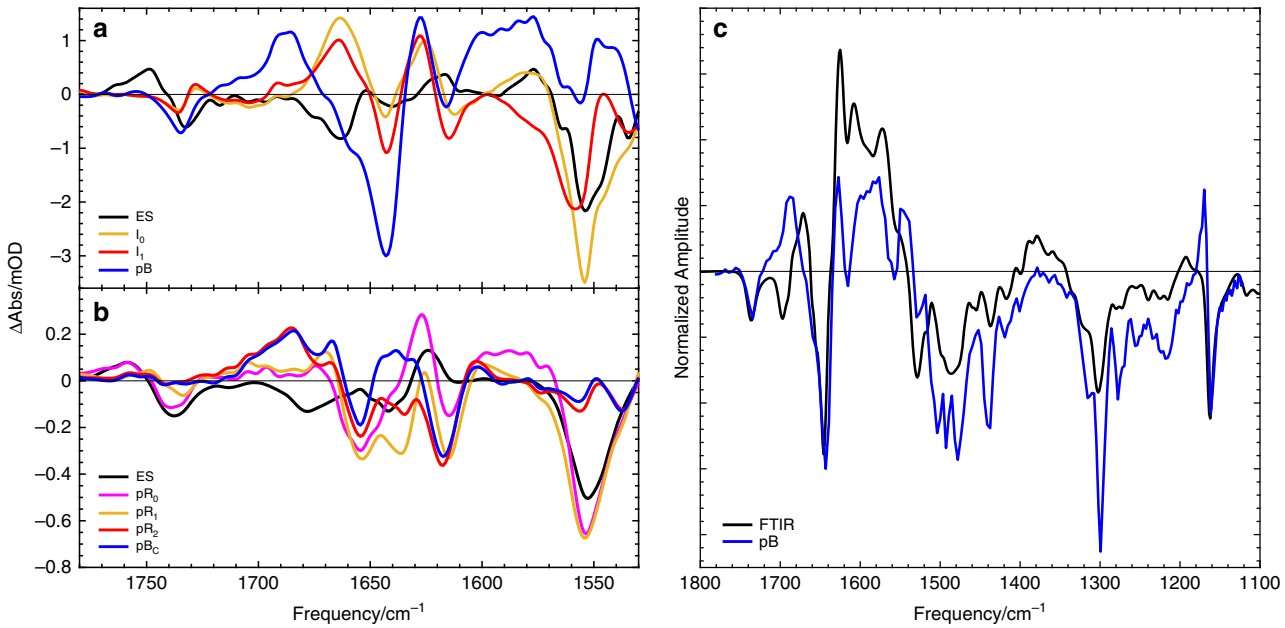

**Fig. 7 Species-associated-difference spectra (SADS) in the mid-IR spectral range.** The SADS have been resolved from fitting the target model shown in Fig. 4 to the experimental data, and display the bleached ground state as negative feature and product state absorption as positive feature. **a** SADS of $PYP_S$, **b** SADS of $PYP_C$. **c** An accumulated PYP light-minus-dark difference FTIR spectrum and the SADS of the pB state of PYP in solution (also shown in **a**). Absorption changes in the 1740–1760 $cm^{-1}$ region report on the hydrogen bond interaction between Glu46 and the phenol ring of pCa: higher frequency indicates weakened bond, lower frequency a stronger bond; Absorption changes in the 1690–1630 $cm^{-1}$ region report on carbonyl stretches with a similar effect of a hydrogen bond on its frequency. The 1555 $cm^{-1}$ band has been assigned to pCa C = C stretch and phenol ring modes[50,51,53], see also Supplementary Table 1.

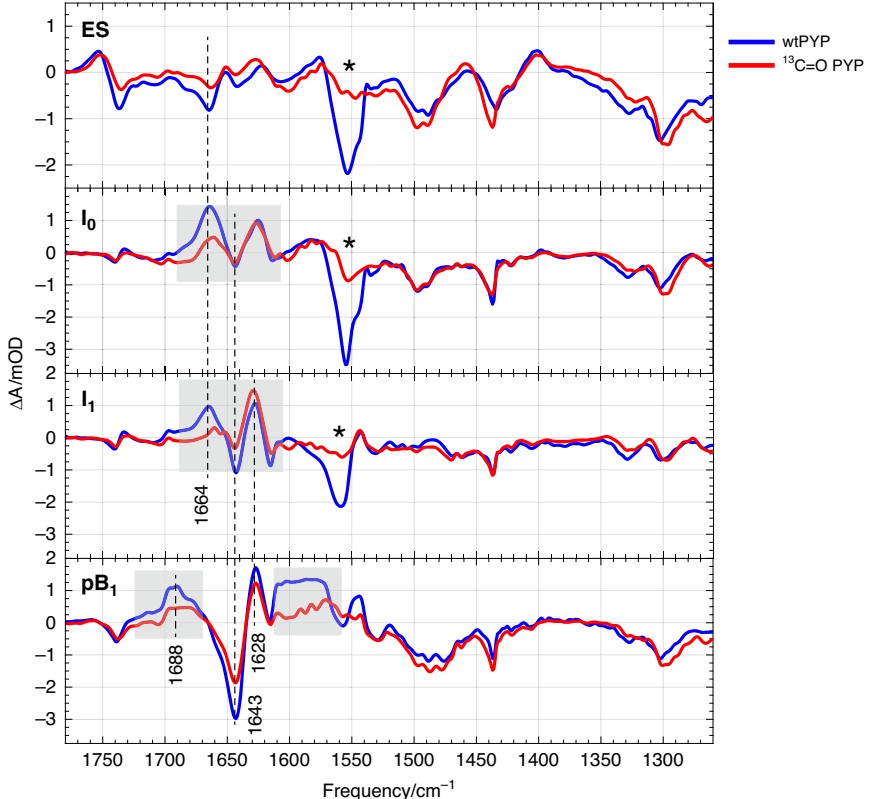

**Fig. 8 Mid-IR SADS of wild type and isotope-labeled PYP at position C9.** WT-PYP (blue) and $^{13}C = O$ - PYP (red) SADS of the ES, $I_0$, $I_1$, $pB_1$ states are shown in the region between 1780–1250 $cm^{-1}$. In the region 1540 to 1250 $cm^{-1}$ the labeled PYP spectrum was scaled by a factor 0.5 for better comparison with the unlabeled spectra, whereas from 1780 to 1540 $cm^{-1}$ no scaling was performed. The differences induced upon labeling of the carbonyl of pCa of PYP with $^{13}C$ are highlighted with colored boxes in the graph and systematic losses around 1555 $cm^{-1}$ bleach is shown by a star (*) symbol.

the signaling state can be observed when comparing the $PYP_S$ $pB_1$ spectrum with a difference spectrum collected in a steady-state FTIR spectrometer using light-on/light-off accumulation (Fig. 7c).

## Discussion

We conclude that our transient spectroscopic measurements on $PYP_C$ are remarkably consistent with the photocycle events as observed in the TRC experiments of Schotte[2], Jung[3] and Ten Boer et al.[11]. Nevertheless, these dynamics differ from PYP dissolved in solution from the initial stages of the photocycle. The question arises of which transient structures the $I_0$ and $I_1$ intermediates in solution represent. Given the upshifted pCa C=O band in $I_0$, indicating a broken hydrogen bond with Cys-69 following isomerization about the C7=C8 double bond, it seems reasonable that this intermediate shares many structural properties with the $pR_1$ intermediate observed in TRC and in the transient IR experiments. It should be noted, however that in $pR_1$, the hydrogen bonds with Glu-46 are relaxed[2] (Fig. 7b), while in $I_0$ they are strengthened (Fig. 7a), which implies that the position of pCa relative to its binding pocket must be slightly different under the two conditions. Even so, $I_0$ likely represents a fully isomerized, quasi-$pR_1$ state formed immediately in 600 fs, while in the crystal the concurrent $pR_0$ state represents an incomplete isomerization that remains stalled for 0.3 ns before isomerization is completed. We conclude that a $pR_0$-like intermediate is never formed in solution. In the crystal, the $pR_2$ intermediate evolves from $pR_1$ through a *syn-anti* rotation of the C8-C9-S bonds to relieve steric strain of the latter pCa structure[2,3]. Such a process is likely to occur in the solution phase as well, so the $I_1$ intermediate probably represents a quasi-$pR_2$ structure. Such *syn-anti* rotation occurs significantly faster in solution (1 ns) than in the crystal (18 ns). Again, we note that the patterns of hydrogen-bond relaxation or strengthening with Glu-46 are reversed between crystal and solution for $I_1$ and $pR_2$ (Fig. 7a, b).

An overall picture emerges, where in solution, a rapid succession of isomerization and rotary motions finally result in a structurally relaxed, red-shifted pCa chromophore on a 1 ns timescale. In the crystal, similar but not identical motions take place that are significantly delayed with respect to solution and where $pR_0$ represents a temporarily arrested isomerization. Considering the early onset of the diverging structural dynamics and their localized nature, the confining effects of crystal packing may be less likely the cause of our observations. In the Yeremenko study[19] the effect of the presence of other solutes in the crystallization buffer, i.e., PEG 2000, which is used as a precipitant in growing crystals, was investigated. Increasing the PEG2000 concentration to the limit of solubility (200 g/L) was found to have no effect on $PYP_S$ dynamics. A difference in pH (8 in solution versus 6 in the crystal) also cannot be the cause, as the $PYP_S$ dynamics until 5 ns have been shown to be independent of pH in a range between 5 and 10[31], which we confirmed here by recording the dynamics at both pH 6 and 8 (see Supplementary Fig. 1). The decreased level of protein hydration in the crystalline phase with respect to that in solution may be the more likely cause for the observed differences between crystal and solution. Overall lower hydration in the crystalline state may lead to a lower water concentration throughout the protein scaffold including the pCa binding pocket and altered electrostatic interactions. The influence of dehydration on the later part of the photocycle of PYP has been studied in films with low hydration levels[56] and cell-mimetic environments[57]. Dehydration was found to alter both the mechanism and the kinetics of the later parts of the photocycle significantly[56,57]. On earlier timescales, the isomerization quantum yield and rate have been found to be

sensitive to the arrangement of water molecules inducing altered hydrogen bond interactions in the PYP active site[28]. However, the molecular basis of the different ultrafast parts of the photocycle of $PYP_C$ and $PYP_S$, whether this is dehydration, altered viscosity or confinement by the crystal lattice remains to be decided and it may well be a mixture of these factors.

Another effect often considered when comparing spectroscopic and TRC results, is the excitation density, which is typically high in TRC experiments, to achieve a high population of the photocycle intermediates. A recent report focused on the comparison of 0.25–10 ps dynamics of bacteriorhodopsin in membranes and crystalline form, investigating the effect of the high power excitation regimes employed in TRC experiments on the dynamics[14]. Here, we used a low excitation density for both cases, to avoid multi-photon processes. Remarkably, the spectroscopic results on crystalline PYP could be well linked to those of the various TRC experiments, suggesting that in PYP high(er) excitation densities overall lead to a very similar photocycle. Overall, these previous accounts and ours above strongly suggest that transient crystallographic and spectroscopic techniques are highly complementary and most effective when applied in a symbiotic fashion in the context of resolving of protein dynamics with varying sample treatment, in order to relate to the dynamics in-vivo.

## Methods

**Protein expression and purification.** wtPYP and $^{13}C=O$ – labeled PYP were produced by reconstituting apo-PYP with the 1,1′-carbonyldiimidazole derivative of *p*-coumaric acid chromophore[58,59] For experiments in aqueous solution the reconstituted holoproteins were purified by using an Äkta FPLC system (GE Healthcare) in two subsequent steps: Ni-affinity chromatography and anion exchange chromatography, respectively.

Reconstituted PYP in cell free extract[53] was loaded on two Ni-affinity columns (HisTrap FF, GE Healthcare), washed with 10 column volumes of buffer A (20 mM $Na_2HPO_4$, 20 mM imidazole, 150 mM NaCl pH =7.5), and eluted with a gradient with buffer B (20 mM $Na_2HPO_4$, 500 mM imidazole, 150 mM NaCl, pH = 7.5) from 0 to 100% in 10 column volumes. The fractions containing PYP were then dialyzed overnight against 20 mM Tris pH = 8.0 and further purified with anion exchange chromatography (HiTrapQ HP columns) with a gradient of buffer A (20 mM Tris, pH = 8.0) and buffer B (20 mM Tris plus 1 M NaCl, pH = 8.0), after filtering of the protein solution (0.2 μm filter). After washing 10 column volumes with buffer A, a gradient of 0 to 12.5% B in 10 column volumes and subsequently a gradient of 12.5 to 20% B in 3 column volumes was used. After that the column was regenerated with 100% B.

To crystallize PYP an extended purification protocol was used: The purified protein was first dialysed to 20 mM Tris buffer pH = 8 and concentrated to >20 mg/ml with a 5.000 MW spin concentrator. The N-terminal poly-histidine tag of the protein was then removed via overnight cleavage with Enterokinase (1.000:1 (w/w); Sigma-Aldrich) at 37 °C. Non-digested protein was removed via Ni-affinity chromatography (see above) and the holo-PYP collected from the flow-through fractions of the latter step was subjected to anion exchange chromatography (using a HiTrapQ HP column) with a gradient (see above) of buffers A (20 mM Tris pH = 8.5) and B (20 mM Tris, 1 M NaCl, pH = 8.5), again after filtering of the protein (see above). The fractions from this column with the highest PYP content were then checked for the dynamics of photocycle ground-state recovery at 464 nm, and for purity on SDS_PAGE, filtered, pooled, and concentrated >25 mg/ml with a 5000 MW spin concentrator.

The final purification step to produce PYP suitable for crystallization experiments was a gel filtration step with a Superdex 75 column in 100 mM MES, pH = 6.5 with ascending flow of 0.5 ml/min and the protein in the fractions with the highest purity was concentrated to ~25 mg/ml in 100 mM MES, pH = 6.5 with a 5000 MW spin concentrator.

Accordingly, the purified PYP protein was used for spectroscopy in aqueous solution without prior removal of the genetically introduced N-terminal poly-histidine tag, in 20 mM Tris buffer, pH = 8. For $^{13}C=O$ – labeled PYP, the purification of $^{13}C$- over $^{12}C$-labeled protein was checked with mass spectroscopy and found to be 100%. Both unlabeled and labeled samples were pipetted in between two $CaF_2$ windows (25 mm diameter, 2 mm thickness) separated by a 10 μm Teflon spacer, resulting in samples with $OD_{446} = 0.8$ /10 μm for both the visible and mid-IR experiments. All experiments were performed at room temperature. See Supplementary Fig. 3 for the steady-state UV-vis absorption spectrum.

**PYP crystallization protocol.** Crystalline PYP in space group $P6_5$ was prepared using the vapor diffusion method drop method according to published protocol[19].

In short, 4 μl of concentrated PYP solution (27.5 mg/ml) was combined with equal volume mother liquor (100 mM MES buffer pH 6.5, 40% PEG 2000) on a circular microscope coverslip. This suspension was carefully fixed over a compartment of a 24 well plate containing 1 ml mother liquor and sealed with vacuum grease. The crystallization proceeded for 6 days at 21 °C and crystals were then harvested using a micropipette. The resulting product was washed with fresh mother liquor and stored at 4 °C. For transient absorption measurements, crystals were loaded between two $CaF_2$ windows without a spacer and crushed for optimal transparency, similar to the procedure in ref. [60]. A minimal amount of scatter was observed cf, Fig. 1. The $OD_{446}$ was ~0.8 but varied widely within the sample. A $D_2O$ mother liquor was used for the mid-IR experiment to reduce background buffer absorption.

**Transient electronic absorption spectroscopy**. Dual regeneratively-amplified laser systems (Legend and Libra, Coherent, Santa Clara, CA) seeded by a common Ti:Sapphire oscillator generated <50 fs pulses at 800 nm with 1 kHz repetition rate[61-63]. The Legend was used as a pump source for an optical parametric amplifier (OPerA Solo, Coherent) to produce 475 nm light with 15 nm bandwidth FWHM. A broadband supercontinuum probe beam was generated by focusing ≈5% of the Libra output onto a translating $CaF_2$ plate. The pump (140 nJ) and probe beams were temporally and spatially overlapped on the sample. The pump beam was focused to a spot with a FWHM of 250 μm, and the probe beam to 125 μm. The probe was spectrally dispersed and detected with a multichannel detection system (Entwicklungsburo Stresing) comprised of a 1024 pixel back-thinned FFT-CCD detector (S7030-1006, Hamamatsu). The transient absorption signal was measured in situ by modulation of the pump beam with an optical chopper at one-half the laser repetition rate. An excitation intensity of 400 nJ per pulse was used for both samples. In combination with the off-center excitation wavelength of 475 nm this resulted in an excitation density of ~5% excited proteins. This value was carefully chosen to avoid multi-photon processes and/or photoionization[20,60]. The polarization of pump and probe beams was under the magic angle. In combination with the crushing procedure of the crystals, this minimizes possible photo-selection effects in the crystals.

**Transient mid-IR absorption spectroscopy**. Transient absorption data in the mid-IR spectral region were recorded with a similar setup utilizing dual femtosecond Ti:Sapphire amplifier systems seeded by a common oscillator (MaiTai, Spectra Physics). The first (1 kHz, Hurricane, Spectra Physics) produced ≈85 fs pulses at 800 nm with 0.6 mJ pulse energy which was used to pump an OPA (TOPAS, Light Conversion) tuned to 475 nm for electronic excitation. The output of the second amplifier (1 kHz, Spitfire Ace, Spectra Physics) pumped a separate OPA (TOPAS-C, Light Conversion) to produce near-IR signal and idler beams which then underwent subsequent difference frequency mixing in $AgGaS_2$ to yield ≈100 fs mid-IR pulses from 1–10 μm with ≈200 cm$^{-1}$ bandwidth FWHM. The IR probe beam was focused on the sample with a 10 cm $CaF_2$ lens (125 μm spot diameter) and spatially and temporally overlapped with the excitation beam (250 μm spot diameter) in the sample plane. The emerging probe beam was dispersed in a grating spectrograph and detected with a liquid nitrogen-cooled 64 element Mercury Cadmium Telluride photodiode array (Infrared Associates). The setup was contained in a box continuously purged with dry air to minimize ambient water vapor absorption. An excitation intensity of 1 μJ per pulse was used for both samples. In combination with the off-center excitation wavelength of 475 nm this resulted in an excitation density of ~10% excited proteins. This value was sufficiently low to avoid multi-photon processes and/or photoionization[20]. The polarization of pump and probe beams was under the magic angle. In combination with the crushing procedure of the crystals, this minimizes possible photo-selection effects in the crystals.

**Data collection and analysis**. The transient absorption data was collected as previously described[61,64]. In short, probe delays up to 6 ns were achieved with an optical delay line and long pump-probe delays (from 12 ns to 500 ms) by systematic adjustment of the pump laser Pockels cell timing. The solution sample was refreshed using a home-built Lissajous scanner, whereas the crystalline sample was continuously scanned in a linear pattern to prevent photodamage and allow for adequate dark state recovery during data collection. The integrity of the sample was monitored by measuring its linear absorption spectrum before and after data collection. For the mid-IR experiments, data was collected in the range of 1540–1780 cm$^{-1}$. Labeled and unlabeled samples were measured consecutively and repeated to ensure reproducibility. Calibration of the spectrometer was done using polynomial fitting of vibrational bands of known reference samples. This procedure had an uncertainty of 4 cm$^{-1}$, therefore we used the results from the FTIR measurements to correct the time-resolved IR spectra by applying up to 4 cm$^{-1}$ shifts. The transient absorption data were analyzed by global analysis using the Glotaran software package[65]. Details of this method and analysis of the data are described in the section below.

**Target analysis**. To extract the pure species-associated spectra, we applied a target model providing a physical description of the photocycle of PYP. Pump-dump-probe studies, previous transient absorption studies[23,24,26-28] were used to confine

our target model. With this method, we can extract the characteristic spectra of all photocycle states, namely, contributions from the excited state (ES), primary photoproduct ($I_0$), and subsequent intermediates. The photocycle scheme and the corresponding parameters for the kinetic model are shown in Fig. 4.

**Reporting summary**. Further information on experimental design is available in the Nature Research Reporting Summary linked to this paper.

## Data availability

Source data are provided as a Source Data file. Other data are available from the corresponding author upon reasonable request. Source data are provided with this paper.

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

## Acknowledgements

This work was supported by the Dutch organization for scientific research NWO, through the divisions of Earth and Life Sciences (ALW) and Chemical Sciences (CW), through an Open Programme grant to M.L.G., Medium Large Investment grants to M.L.G., K.J.H. and J.T.M.K. P.K. and J.T.M.K. were supported by a NWO-VICI grant. The authors would like to thank Dr. S. Yeremenko for reuse of the PYP-TA datasets.

## Author contributions

P.K. and E.A. conducted the experiments, J.W. programmed the transient absorption setup, J.A. isolated and purified the proteins, I.H.M.v.S. and M.L.G. analyzed the results, M.L.G., E.A. and P.K. wrote the manuscript, E.A., J.T.M.K. and K.J.H. provided input to analysis and manuscript, K.J.H. and M.L.G. conceived and supervised the project. All authors have given approval to the final version of the paper.

## Competing Interests

The authors declare no competing interests.
