## [Peer Review File · Nature Communications]

Reviewers' comments:

Reviewer #1 (Remarks to the Author):

The manuscript by Konold et al describes a comparison of the photocycle kinetics of Photoactive Yellow Protein (PYP) measured on solution samples and crystals, respectively. PYP is a popular model system that has been studied using numerous approaches, including time-resolved crystallography. It is well known that the dynamics and kinetics of crystalline molecules can differ from those in solution. Crystal contacts and other steric constraints can affect mobility; the composition (pH, ionic strength, ...) of the crystal containing solution can affect chemical reactivity. Thus, previous studies exist that compare the PYP photocycle in solutions and crystals. Yemeko et al (Biophys J, 2006) performed studies with microsecond time resolution and noticed distinct differences in both time constants and pathways. Importantly, the two samples were measured at the same pH; the photocycle of PYP is highly pH sensitive (Hendriks & Hellingwerf, JBC 2009). Recently, the photocycles of PYP crystals and solutions were compared on the ultrafast time-scale (Hutchison et al Chem Phys Lett, 2016). Again, differences were noted, but the focus was not a comparison of dynamics/kinetics but to determine the parameters maximizing photocycle yield of the long lived pB intermediate, and by extension of intermediates occurring on the ultrafast time scale. An important variable was the photoexcitation power and wavelength.

The manuscript by Konold et al compares the photocycle of crystalline and PYP using low power excitation, from sub-ps to seconds. The major finding is that large differences are observed on all time scales. Unfortunately, the crystal and solution experiments were performed at very different pH values (pH 8 solution, pH 6.5 crystals). It is therefore unclear whether the observed differences are caused by the pH, the lattice effects, or both. The authors ignore the pH effect and specify that the differences are due to lower hydration in the crystal, without giving any justification/proof for this statement (which is at most a hypothesis). Dehydration in films almost certainly differs from those in crystals.

Thus, there are three major issues with this manuscript:

- 1) The biggest: two important variables were changed in the experiments (pH, sample state (crystal/solution)), but only the latter was discussed in causing the differences.
- 2) The bottom line: "Time-resolved crystallography cannot be blindly extrapolated to in vivo function". This is never the case for crystal structures and well known.
- 3) The differences in kinetics between solution and crystals are attributed to differences in hydration without offering any proof for this.

Minor points:

Page 3, line 2: wrong references.

Page 3 "earliest time-resolved (and static) crystallographic: (and static) is wrong.

Page 4 "as observed in the majority of the" .. which ones are not?

Page 14: temperature for spectroscopic measurements?

Page 15: Suspended drop -> hanging drop

Page 15: crystal size? Spacer thickness? Approach similar to the one described by Hutchison et al Chem Phys Lett, 2016? OD of crystalline sample? What about scattering contribution?

Laser spot diameter of 250 um: FWHM? $1/e^2$?

Reviewer #2 (Remarks to the Author):

Konold et al. studied photocycle of Photoactive Yellow Protein (PYP) in crystal and solution by time-resolved UV-Vis and infrared absorption spectroscopy, and compared its photocycles in a very wide time range from 100 femtosecond to second. On the basis of species-associated -difference-spectra obtained with a target model, they concluded that the photocycle kinetics and the properties of the intermediates in the crystal significantly deviate from those in solution, and

argued that the results of femtosecond time-resolved crystallography (TRC) cannot be blindly extrapolated to consider the protein *in vivo*.

TRC is certainly an extremely powerful tool to study ultrafast structural dynamics of photoreceptor proteins and hence attracts huge attention these days. Recently, however, people are aware of several serious problems of the TRC experiments such as the unwanted effect induced by the violent excitation condition. In this paper, the authors discussed the effect of confinement in crystal lattice and lower hydration environment of the sample used for TRC. Because they are inevitable in TRC experiments, I think that the authors address a very important problem. Nevertheless, although I feel that the essential points of their arguments would be true, this paper has a number of serious problems as I write below. Therefore, I do not recommend publication of this paper, not only in *Nature Communication* but also in any other journals.

1. A very serious problem of this paper is that the authors made most of their discussions based on the species-difference-spectra (SADS). SADS are very dependent on the model used in the analysis, and they can readily be changed with the change of the model. The authors used a very complicated model (given only in SI). This scheme has already been published but I do not think that this model is widely accepted in the community. It is fine to show and discuss SADS in the paper (with sufficient explanation). However, the essential discussions should be made for the raw time-resolved spectra which they do not show in the paper. This is critically important for readers to figure out which arguments are based on the time-resolved data themselves and which interpretations are model-dependent. Furthermore, because the author did not describe the details of their analysis even in Supporting Information, it is not clear which parameters they obtained with fitting and which values they used as fixed values. In fact, the authors discussed the difference in the time constant of the dynamics between crystalline-phase and solution phase, but I cannot follow how they reached this conclusion in a logical way. In addition, I believe that the authors also need to show the steady-state UV-visible and infrared spectra of PYP.

2. One of the main arguments in this paper is that the structure of the I₀ intermediate (in solution) is different from pR₀ intermediate (in crystal), and that the I₀ intermediate has the fully isomerized chromophore. This argument is based on the CO stretch frequency of the chromophore that was evaluated for time-resolved IR absorption SADS. The authors are not aware but a recent femtosecond impulsive stimulated Raman (TR-ISRS) study of PYP in solution clearly showed that the CO stretch frequency of I₀ is quite low, indicating that I₀ state has a distorted *cis*-form which is similar to pR₀ intermediate in the crystal (*Nat. Chem.* 2017, 9, 660). Considering that TR-ISRS selectively detects the vibrations of the chromophore with the resonant Raman enhancement and that the S/N of the reported TR-ISRS spectrum is high, the results of the TR-ISRS study are very trustworthy.

3. Another serious problem of this paper is that the authors do not provide sufficient descriptions and data in the main text. Some information is given in Supplementary Information but they did not make sufficient efforts to write "understandable" SI. Actually, I have an impression that the authors wrote this paper in a too-concise way, trying to publish it in a high-impact journal. Unfortunately, this makes the quality of the scholarly presentation of this paper low. This is a bad aspect of the trend of these days.

Reviewer #3 (Remarks to the Author):

This paper presents the first (to my knowledge) direct and detailed comparison of the dynamics of a photoactive protein in both its solution and crystalline phases. This is a key objective which will underpin the assessment of the reliability and usefulness of the growing number of femtosecond time resolved protein crystallography experiments – which are most frequently applied to photoactive proteins. The target here, PYP, has been thoroughly studied in solution, by ultrafast crystallography and by QM/MM calculation. The very important observation is made that the PYP

photocycle dynamics (and yields) are influenced by crystallization across the entire time range from sub-picosecond to millisecond. Consequently it must be concluded that crystallization modifies even small scale (chromophore localized) dynamics, which is an important conclusion. Significantly, the measurements of PYPC kinetics overlap quite nicely with the crystallography data, which is a second useful result, given current concerns about the very levels of high excitation used in fs crystallography. Thus this paper contains important results in an area which is currently one of great importance, and should be published in NCOMM. A few suggestions for revision follow.

The presentation of the observations and the data analysis are too mixed. The fact that significant differences are observed on all time scales is more important than identifying and characterizing kinetic intermediates. It is unlikely that the new intermediates found in the crystalline state are physiologically relevant. Thus the kinetic data (e.g. figure 1, left) should be accompanied on the right side of the figure by the experimental time resolved spectra, rather than the model dependent SADS. This will better illustrate the experimental differences in all states as well as at all times. The wavelength resolved data are presented in the supporting information, but these only allow an assessment of the fit quality – the spectra themselves will give the most striking evidence of the effect of packing.

The subsequent characterization of the 'photocycles' (current Figure 2) following kinetic analysis to yield the SADS should follow the presentation of the experimental data. Were the number of intermediates simply determined from the minimum number that yielded an acceptable fit? If so could the crystalline state be exhibiting a broader distribution of relaxation rates in the early nanosecond time range – more a relaxation continuum than a number of additional intermediates – or are these intermediates truly distinct? Similarly, the actual ES lifetimes are revealed in the full analysis (Figure S5) to be a sum of three different ES decays; are the PYPS/C differences observed a consequence of a different distribution among the three ES? These details of the analysis should be presented, although they are probably better discussed in supplementary information and referenced from the main text.

One very important result was that the same kinetics for PYPS fit both the optical and IR data. Was the same true for PYPC? If not what were the differences?

The use of 'mesoscopic' and 'mesoscopic context' throughout is distracting. It suggests some intermediate lengthscale dependent results. However, everything in both solution and crystal is essentially described at the single molecule level – the effect of crystallization on the kinetics is discussed (very reasonably) in terms of packing and hydration levels.

Minor matters

In the abstract, it does not seem necessary to specify 'during catalysis'.

The claim in the opening paragraph for sub-Angstrom spatial resolution for IR measurements is perhaps an exaggeration.

When comparing the PYPS/C SADS it may be helpful to give on the figure the rate connecting them as well as the state label.

Top p9, is there some structural feature that lead to the conclusion 'more hydrophobic' rather than 'less H-bonding'?

Top p10, is 'weakened' better than 'relaxed'?

In materials and methods please give the OD of the samples used.

In summary, this is an important paper making a novel and important contribution to the ongoing assessment of the reliability and significance of femtosecond crystallography, and should be published in NCOMM.

In response to the comments by the reviewers we have submitted a revised version of the manuscript in which we have:

- a) Performed extra vis-vis pump probe experiments at pH6 and show that their outcome exactly matches those obtained at pH = 8.
- b) Show raw absorption difference spectra and more time traces in figure 1.
- c) Made a distinction between raw data and analysis (cf figs 1 and 2)
- d) Added a comparison of the pB SADS to a steady state FTIR PYP difference spectrum (fig. 3)
- e) Added midIR time traces to figure 3
- f) Moved the C13 isotope labeled results into the main text (figure 4)
- g) Extended the text to address all questions of the reviewers.

Below we provide our replies to the comments of the reviewers in black and repeat the reviewers' remarks in blue.

Reviewer 1 comments: It is well known that the dynamics and kinetics of crystalline molecules can differ from those in solution. Crystal contacts and other steric constraints can affect mobility; the composition (pH, ionic strength, ...) of the crystal containing solution can affect chemical reactivity. Thus, previous studies exist that compare the PYP photocycle in solutions and crystals. Yeremeko et al (Biophys J, 2006) performed studies with microsecond time resolution and noticed distinct differences in both time constants and pathways. Importantly, the two samples were measured at the same pH; the photocycle of PYP is highly pH sensitive (Hendriks & Hellingwerf, JBC 2009). Recently, the photocycles of PYP crystals and solutions were compared on the ultrafast time-scale (Hutchison et al Chem Phys Lett, 2016). Again, differences were noted, but the focus was not a comparison of dynamics/kinetics but to determine the parameters maximizing photocycle yield of the long lived pB intermediate, and by extension of intermediates occurring on the ultrafast time scale. An important variable was the photoexcitation power and wavelength.

The reviewer appears to refer here to observations that the dynamics and kinetics of crystalline molecules can differ from those in solution on long time scales, when large structural changes (have to) take place. Our manuscript reports the fact that from the earliest, sub-picosecond, time scale onwards, the dynamics in crystals differs from that in solution. This effect has not been studied and reported in the literature for any protein, to our knowledge. The data from the study by Yeremenko, included in our report to obtain a more reliable estimate of the dynamics on the >100 microsecond time scale, report on the long time scale dynamics. The Hutchison paper does not report kinetics of crystalline molecules as compared to those in solution. The focus of that paper is on maximizing photocycle yield of the long lived pB intermediate, varying photoexcitation power and wavelength. We could not find any data in this publication showing a difference in dynamics between crystal PYP and PYP in solution. We are happy to include a reference to this paper in the text, as ref # 62. The reviewer further refers here to Hendriks and Hellingwerf JBC 2009, where the recovery of the photocycle, i.e. the deprotonation of the p-coumaryl chromophore, refolding of the protein, and chromophore re-isomerization were found to be pH sensitive. This occurs on the timescale of 1-30 seconds and is irrelevant for the discussion of the ultrafast experiments in our current manuscript.

The manuscript by Konold et al compares the photocycle of crystalline and PYP using low power excitation, from sub-ps to seconds. The major finding is that large differences are observed on all time scales. Unfortunately, the crystal and solution experiments were performed at very different pH values (pH 8 solution, pH 6.5 crystals). It is therefore unclear whether the observed differences are caused by the pH, the lattice effects, or both. The authors ignore the pH effect and specify that the differences are due to lower hydration in the crystal, without giving any justification/proof for this statement (which is at most a hypothesis). Dehydration in films almost certainly differs from those in crystals. Thus, there are three major issues with this manuscript:

1) The biggest: two important variables were changed in the experiments (pH, sample state (crystal/solution)), but only the latter was discussed in causing the differences.

Dynamics on the early time scale have been reported to be insensitive for the pH in a range of 5-10, by Stahl, ... and Hellingwerf in (Biophys. Journal 101, p.1184-1192, 2011). We have in the revised version included a reference to this paper. In addition, to underscore this observation further, we have performed an additional experiment on PYP in solution at pH 6 and overlaid the traces with those collected at pH 8 in figure S1. The dynamics at pH 6 and pH 8 are found to be the same.

2) The bottom line: "Time-resolved crystallography cannot be blindly extrapolated to in vivo function". This is never the case for crystal structures and well known.

The reviewer may have this opinion, but this has so far not been proven by experiments. We are of the opinion this is not 'common knowledge', as to date, in TRC literature no attention has been given to this issue. In fact, whereas for large structural changes that occur later in the photoreaction, awareness certainly exists, the common expectation in the field would be that early time scale dynamics (femtoseconds to nanoseconds) are the same under crystalline and solution conditions.

To do justice to this discussion, we modified the last sentence of the abstract and the last sentences of the discussion into: *This illustrates that even ultrafast TRC results cannot be uncritically extrapolated to in vivo function, and that parallel spectroscopic experiments on proteins in crystalline and solution state can help to identify structural intermediates in native condition.*

3) The differences in kinetics between solution and crystals are attributed to differences in hydration without offering any proof for this.

The main point of the article is that, under the same excitation conditions, the kinetics of PYP in solution differ from those in crystals, from the earliest time scale onwards. We ruled out a pH effect (now, thanks to the reviewer, more clearly) and discuss both hydration and confinement as possible causes. We cite studies from the literature that make hydration in this case more likely to be the cause than confinement.

Minor points:

Page 3, line 2: wrong references.

We reformulated this sentence to more specifically address the topics represented by references 1-7.

Page 3 "earliest time-resolved (and static) crystallographic: (and static) is wrong.
We deleted '(and static)'.

Page 4 "as observed in the majority of the" .. which ones are not?

Here, we meant that not all TRC studies had the same time resolution, and that in the TRC studies similar results have sometimes been modeled differently, not that our results differ from some of the studies. We changed this sentence to "as observed in the TRC experiments".

Page 14: temperature for spectroscopic measurements?

All experiments were performed at room temperature, we inserted this at the end of section Protein Expression and Purification.

Page 15: Suspended drop -> hanging drop

In ref 18 (now 19), Yeremenko et al, this method is referred to as vapor diffusion method, therefore we replaced suspended drop with this term in the section PYP Crystallization Protocol.

Page 15: crystal size? Spacer thickness? Approach similar to the one described by Hutchison et al Chem Phys Lett, 2016? OD of crystalline sample? What about scattering contribution? Laser spot diameter of 250 um: FWHM? $1/e^2$?

The sample was prepared in a similar fashion to Hutchison 2016, i.e. no spacer, squeezed between two windows. There was no obvious scatter contribution. Laser spot: ~250 um FWHM. Crystal OD: 0.3 for uvvis, 0.8 for midIR (varied widely within the sample). This information has been added to the Materials and Methods.

Reviewer 2 starts by remarking that 'I think that the authors address a very important problem. Nevertheless, although I feel that the essential points of their arguments would be true,...' which we very much appreciate, and then raises three points:

1) A very serious problem of this paper is that the authors made most of their discussions based on the species-difference-spectra (SADS). SADS are very dependent on the model used in the analysis, and they can readily be changed with the change of the model. The authors used a very complicated model (given only in SI). This scheme has already been published but I do not think that this model is widely accepted in the community. It is fine to show and discuss SADS in the paper (with sufficient explanation). However, the essential discussions should be made for the raw time-resolved spectra which they do not show in the paper. This is critically important for readers to figure out which arguments are based on the time-resolved data themselves and which interpretations are model dependent.

Crucially, all our conclusions are based on the observation that the raw kinetics in PYP_C and PYP_S are different. To that end, figures 1, S1 and S2 showed 40 raw time traces that all show dynamics different for PYP in crystalline form than for PYP in solution. Given the extended space we have now in the format for Nature Communications (as compared to that of Nature) and the request of the reviewer to see more raw time-resolved spectra, we have remade figure 1 to include a set of 13 time-gated spectra for both PYP_S and PYP_C, and inserted an extra pair of time traces. In the paragraph headed 'Transients in the UV-visible spectral region', we aid the reader in the interpretation of these difference spectra and time traces, and we added the sentence there: 'For instance, the dynamics on the (sub-) picosecond time scale appear different, the yield of photoproduct formation in PYP_C is lower than in PYP_S, and dynamics in PYP_C on a 10-20 ns time scale is observed that is absent in PYP_S.' We have added four sets of the transients recorded in the midIR spectral region to Figure 3, and discuss how they track key events in the photocycle. Of course they also illustrate that similarly, in the midIR the dynamics of PYP in solution are different than in the crystalline state.

After quantitatively appreciating that the raw time-gated spectra and time traces of PYP_S and PYP_C are different, we have to fit and analyze the data. The excited state decay dynamics of PYP has been extensively shown to be multi-phasic, both by transient absorption, fluorescence upconversion and fluorescence streak camera experiments. Furthermore, pump-dump-probe experiments have demonstrated the involvement of a ground state intermediate in the excited state decay. Finally,

visible pump – mid infrared probe experiments have provided a reliable estimate of the quantum yield of photoproduct (I0, I1) formation. This has culminated in the description of the photocycle of PYP in terms of a target model, which has been extensively used and described in at least 15 earlier publications, of which we reference a few of the most essentials ones, refs 23-25.

Though the target model displayed in Figure S6 appears complicated, after the dynamics of the excited state, the model is essentially linear (which is why we represent a simplified linear scheme in Figure 2). A comparison of the new figures 1 and 2 shows that therefore the shape of the SADS correspond directly with the raw data, at the appropriate time points (Note that the longer time scale SADS come from Yeremenko et al, not shown in the time-gated spectra in figure 1). However, the SADS have been scaled to their 'true' value, as the quantum yield of formation of the states during the photocycle has been taken into account. We hope that with this explanation and the added raw data, the reviewer can overcome his/her aversion towards the use of the SADS.

Furthermore, because the author did not describe the details of their analysis even in Supporting Information, it is not clear which parameters they obtained with fitting and which values they used as fixed values. In fact, the authors discussed the difference in the time constant of the dynamics between crystalline-phase and solution phase, but I cannot follow how they reached this conclusion in a logical way. We have provided more detail on the choices made for the target model in the main text, and in figure S5 we have indicated which parameters were fixed, based on earlier studies, and which were free to vary: "The rate constants involving the ES and GSI for PYPS have been fixed from²⁴, all other parameters have been estimated". Further, in the main text we have added the following sentence which links the target analysis to the simplified scheme of fig. 2: 'The initial decay of the ES in PYP_C is fastest, 0.35 ps, but results in only 10% pR₀. A second decay phase in the ES leads to further pR₀ formation in 2.6 ps, resulting in an overall quantum yield of 23%, see fig. S6. For simplicity, we indicate this process with a weighted average time constant of 1 ps in the scheme in Fig. 2. In PYP_S, 29% of I₀ is formed in 0.6 ps and another 2% in 2 ps. We summarize this in the scheme in Fig 2 as: formation of the first cis isomer of pCa, with red-shifted product absorption, is slower in PYP_C than in PYP_S (1 versus 0.6 picoseconds) and is formed with a lower quantum yield (0.23 vs 0.31).' Overall, we have extensively rewritten the section "Photocycle analysis".

In addition, I believe that the authors also need to show the steady-state UV-visible and infrared spectra of PYP. In the revised version we have included the steady-state UV-visible spectrum as fig. S8, and discuss it on p.4: *The spectra of the crystalline samples are slightly red-shifted (by 3–4 nm) with respect to the spectra of PYP in solution, whereas their width and the overall shape are very similar (see Fig. S8), in agreement with earlier observations^{19, 21, 22}.* As steady-state FTIR spectra are not so informative on the state of the protein due to the large water absorption, we have chosen to include a) the results from the isotope labeled experiment in the main text as fig. 3, and b) to include a comparison of the pB₁ difference spectrum collected on the mid-IR setup with a light minus dark steady state absorption difference spectrum collected on an FTIR spectrometer, fig. S7.

2.) One of the main arguments in this paper is that the structure of the I0 intermediate (in solution) is different from pR0 intermediate (in crystal), and that the I0 intermediate has the fully isomerized chromophore. This argument is based on the CO stretch frequency of the chromophore that was evaluated for time-resolved IR absorption SADS. The authors are not aware but a recent femtosecond impulsive stimulated Raman (TR-ISRS) study of PYP in solution clearly showed that the CO stretch frequency of I0 is quite low, indicating that I0 state has a distorted cis-form which is similar to pR0 intermediate in the crystal (Nat. Chem. 2017, 9, 660). Considering that TR-ISRS selectively detects the vibrations of the chromophore with the resonant Raman enhancement and

that the S/N of the reported TR-ISRS spectrum is high, the results of the TR-ISRS study are very trustable.

In addition to the observation that the kinetics are different, indeed our second most important observation is that the mid-IR spectra of the I_0 and pR_0 intermediates when overlaid are different. By employing isotope labeling of the C=O group of the pCa chromophore we are able to assign the positive product band shifted to higher frequency to the pCa carbonyl, and use that as the basis to make statements about the structure of the pCa chromophore in the various intermediates. We had indeed missed the Tahara study in Nat. Chem. 2017, 9, 660. Although we do not feel it is appropriate to fully discuss the ins and outs of this article here, we would like to note that, due to the broadband Raman excitation wavelength employed, inevitably a mixture of I_0 , GSI and possibly GS is probed in that study even if wavelengths < 500 nm were cut. As this is not taken into account in their analysis, it could very well be that mainly GSI or GS are responsible for the reported CO stretch band frequency. Even so, a different interpretation provided on the basis of a certain set of experiments can of course never mean that we cannot publish another interpretation, as the reviewer appears to suggest, especially because we have performed site-specific isotope labeling to provide a solid basis for our interpretation. We have included a reference to the Tahara study: ref#46, and on p.10 inserted: *'the frequency of pCa carbonyl in the I_0 state has been a matter of contention. To improve the mid-IR spectral assignment, we utilized a ^{13}C isotopic label at position C9 of the pCa chromophore...'*

3. Another serious problem of this paper is that the authors do not provide sufficient descriptions and data in the main text. Some information is given in Supplementary Information but they did not make sufficient efforts to write "understandable" SI. Actually, I have an impression that the authors wrote this paper in a too-concise way, trying to publish it in a high-impact journal. Unfortunately, this makes the quality of the scholarly presentation of this paper low. This is a bad aspect of the trend of these days.

We understand the comment by the reviewer, but for our manuscript we had taken the 'direct transfer option' from Nature to Nature Communications, which explains why the manuscript was written in the concise manner dictated by Nature. We have in the revision of the manuscript provided more figures and arranged them differently, see figures 1, 2,3 and 4, and we provided more description and discussion in the main text. The figures in the SI are now limited to time traces in the visible and mid-IR, zoomed in version of time traces, the UV-vis absorption spectrum and a table containing assignments.

Reviewer 3

This paper presents the first (to my knowledge) direct and detailed comparison of the dynamics of a photoactive protein in both its solution and crystalline phases. This is a key objective which will underpin the assessment of the reliability and usefulness of the growing number of femtosecond time resolved protein crystallography experiments – which are most frequently applied to photoactive proteins. The target here, PYP, has been thoroughly studied in solution, by ultrafast crystallography and by QM/MM calculation. The very important observation is made that the PYP photocycle dynamics (and yields) are influenced by crystallization across the entire time range from sub-picosecond to millisecond. Consequently, it must be concluded that crystallization modifies even small scale (chromophore localized) dynamics, which is an important conclusion. Significantly, the

measurements of PYPC kinetics overlap quite nicely with the crystallography data, which is a second useful result, given current concerns about the very levels of high excitation used in fs crystallography. Thus, this paper contains important results in an area which is currently one of great importance, and should be published in NCOMM. A few suggestions for revision follow.

We are very pleased with this positive assessment.

The presentation of the observations and the data analysis are too mixed. The fact that significant differences are observed on all time scales is more important than identifying and characterizing kinetic intermediates. It is unlikely that the new intermediates found in the crystalline state are physiologically relevant. Thus the kinetic data (e.g. figure 1, left) should be accompanied on the right side of the figure by the experimental time resolved spectra, rather than the model dependent SADS. This will better illustrate the experimental differences in all states as well as at all times. The wavelength resolved data are presented in the supporting information, but these only allow an assessment of the fit quality – the spectra themselves will give the most striking evidence of the effect of packing. The subsequent characterization of the ‘photocycles’ (current Figure 2) following kinetic analysis to yield the SADS should follow the presentation of the experimental data.

In the revised version we have remade figure 1 to display a selection of a double set of 13 raw absorption difference spectra in addition to three time traces, for each PYPS and PYPC, and we have moved the SADS to figure 2, to make indeed a better separation between raw data and analysis. Figure 2 now contains the model for the photocycle, the SADS and the chemical structure of the pCa chromophore.

Were the number of intermediates simply determined from the minimum number that yielded an acceptable fit? If so could the crystalline state be exhibiting a broader distribution of relaxation rates in the early nanosecond time range – more a relaxation continuum than a number of additional intermediates – or are these intermediates truly distinct? Similarly, the actual ES lifetimes are revealed in the full analysis (Figure S5) to be a sum of three different ES decays; are the PYPS/C differences observed a consequence of a different distribution among the three ES? These details of the analysis should be presented, although they are probably better discussed in supplementary information and referenced from the main text.

The PYPS data is fitted with the target model as known and presented in the literature. For the PYPC data it was necessary to include one more state. Including this one more state was sufficient to fit the data, so if indeed there would be physically rather a distribution of states, we expect such a distribution to be narrow. For the three excited states, no distinction can be made whether they are really three separate states, or one state that relaxes and has varying reaction rates over time, or is rather a continuous distribution of states. On p.7 we have included a short discussion on this topic: *Both a sequential model, in which each short-lived excited state relaxes into the next longer-living state, and a parallel model, in which the excited states decay independently, gives an equally good fit to the data. It is therefore up to now unclear whether the different time constants and their different reactivity are representative of a relaxation process in the excited state (i.e. an evolution from a more- towards a less-productive minimum on the excited state surface), or whether they are due to heterogeneity or inhomogeneity, due to small structural differences between proteins*

One very important result was that the same kinetics for PYPS fit both the optical and IR data. Was the same true for PYPC? If not what were the differences?

The same was true for PYPC, we remark this now more clearly at the start of the discussion of the target analysis.

The use of 'mesoscopic' and 'mesoscopic context' throughout is distracting. It suggests some intermediate length scale dependent results. However, everything in both solution and crystal is essentially described at the single molecule level – the effect of crystallization on the kinetics is discussed (very reasonably) in terms of packing and hydration levels.

We have replaced mesoscopic with 'crystal and solution' or an equivalent thereof throughout the text.

Minor matters

In the abstract, it does not seem necessary to specify 'during catalysis'.

We deleted this from the abstract.

The claim in the opening paragraph for sub-Angstrom spatial resolution for IR measurements is perhaps an exaggeration.

We changed this sentence into : *Vibrational spectroscopy offers, after successful assignment, high resolution structural information on interatomic distances and dihedral angles of catalytically active groups.*

When comparing the PYP/C SADS it may be helpful to give on the figure the rate connecting them as well as the state label.

We have now placed the figure with the SADS in one figure with the photocycle, so this connection can be made directly.

Top p9, is there some structural feature that lead to the conclusion 'more hydrophobic' rather than 'less H-bonding'?

In the previous state the 1664 cm⁻¹ position for the carbonyl already indicates that the carbonyl is free from hydrogen bonds, and 1688 cm⁻¹ is very high for a carbonyl frequency. We changed 'more hydrophobic' to 'different environment, possibly more hydrophobic'.

Top p10, is 'weakened' better than 'relaxed'?

We used weakened as opposed to strengthening, but changed it to relaxed.

In materials and methods please give the OD of the samples used.

In M&M we inserted a more elaborate sample description.

In summary, this is an important paper making a novel and important contribution to the ongoing assessment of the reliability and significance of femtosecond crystallography, and should be published in NCOMM.

We thank the reviewer for this assessment.

REVIEWER COMMENTS

Reviewer #1 (Remarks to the Author):

The revised manuscript by Konold et al has undergone substantial changes, is much improved and most of my concerns have been addressed. However, the authors should focus more on what is new. It is well known, or at least published, that the photo cycle differs in solution and crystals. In the latter there are additional states such as pR1 (see e.g. Jung et al) , and the slower time scales deviate quite a bit due to crystal packing constraints.

I realize that the photocycle of PYP is complex and that the different nomenclatures do not help. However, given the focus of the manuscript, differences on the ultrafast time scale in solution/crystal, I feel that it would be more appropriate to show Supplementary Fig. S6 in the main text than the strongly simplified Fig. 2. Even more so as the intricacies of Fig. S6 are described in the main text. It seems to me that the schemes shown in Fig. S6 for solution and crystal are rather similar? If not this needs to be stressed further and more explicitly than currently done. It is a pity that the authors have not made more use of the large amount of space available in Nature Comm.

Along these lines, I find the manuscript rather difficult to read. It is not always clear which sample is being described (crystalline/solution). I understand that the authors want to directly compare features between the two samples instead of describing one sample first, then the other, then the differences but this comes at a price. Not sure it is worth it.

I think a stringent comparison of the ultrafast photocycle in solution and crystals would be interesting. It is important that comparable parameters are used for the experiments. It is therefore very good that the authors also pointed out the reference describing the lack of influence of increasing the PEG concentration on the photocycle kinetics. This limits the reasons for the apparent discrepancy.

There is one further difference that the authors did not mention: The orientation of protein embedded chromophores in solution and crystals differ, they are randomly distributed with respect to the photoexcitation polarization plane in solution whereas photo selection rules likely apply in oriented crystals (see e.g. page 19260 PYP photoactivation; Schotte et al, PNAS). The crystalline pancake approach, making a sample for spectroscopy by squishing a very large crystal between glass plates separated by thin spacers may not be good enough to randomize the orientations of the microcrystalline fragments. This could be avoided by stirring up the squished crystal and letting the crystallites settle. I could understand if the authors did not want to repeat some of the measurements using such a sample, but they should address this problem in the manuscript thoroughly. Or add a disclaimer that this could influence the kinetics in the crystalline state. I take it they use linearly polarized light? (not stated). SFX experiments suffer less from this issue because they use randomly oriented crystals for data collection and most often circularly polarized light.

The authors are likely aware that there are a number of controversies associated with the time-resolved crystallographic data. By no means do Schotte et al and Jung et al agree, see controversy published in Nature Chemistry 2014, this concerns the conformation of the chromophore and the lack of observation of pR1 by Schotte et al. In particular the ultrafast TR-SRX experiments on PYP performed by Schmidt and coworkers used much higher power densities for photoexcitation than considered safe by Hutchinson et al., these experiments were all performed using multiphoton excitation. Photoexcitation was mostly performed at 450 nm, the PYP crystals were grown from salt and not PEG. I am therefore surprised by the statement that "the PYPc results closely follow those by Schotte, Jung and Tenboer [misspelled]. This included... and the 22 % overall pR quantum yield...` Line 175-178. The quantum yield in PYP is wavelength dependent, why would it be the same?

Line 244: "Changes in the region of the ... "this could be checked in the various crystal structures

(e.g. see Table 1, Pande et al), with the caveats mentioned above

Line 250: "Schotte et al reported that in pB, Arg52 switches to an open position.." This is correct. But Arg52 moves because it would otherwise directly overlap with the oxygen of the cofactor in the new pCa conformation in pB. The pCa oxygen is 2.9 Ang from a water molecule (2115) and 2.9 Ang from NE of Arg52. Glu46 forms a hydrogen bond to a water molecule(2056) located at the position of the oxygen atom of dark state pCA (3.2 Ang). The water also forms a hydrogen bond to Tyr42. (Glu-O-pCa: dark2.5 Ang, pR0: 2.7 Ang, pR2: 2.6 Ang); structures from Schotte.

Line 259: " We conclude that our transient measurements on PYP are remarkably consistent with the photocycle elements as observed in the TRC experiment". First, see caveats mentioned above. Second, Pande et al observes full isomerization within 0.6 ps, no further changes take place on the ps time scale. This contradicts the findings described Line 162-165 and 268-270. As mentioned above, there may be many reasons for this discrepancy. But lumping all time-resolved crystallographic experiments together and saying that they agree with the findings of the authors is not adequate for the purpose of this manuscript.

The authors attribute dehydration as the reason for the discrepancies between PYP photocycle kinetics in solution and the crystalline state. Increasing PEG concentrations is used frequently by crystallographers to dehydrate crystals to improve packing. Doesn't the lack of influence of increasing the PEG concentration on the dynamics argue against this? Yeremenko only studied micro to millisecond spectroscopy. In this time range the photocycle differs a lot between solution and crystal anyway. It does not necessarily show that PEG does not influence the kinetics on the fast time scale. This caveat needs to be mentioned. A nice recent example is given by Woodhouse Nature Comm 2020, showing that the presence of ammonium sulfate (used for crystallization) can change the kinetics in a fluorescent protein. Jasper van Thor thinks it is the viscosity of the protein that influences, or at least can influence the ultrafast dynamics upon photoexcitation. He mentions this in one of his publications.

The authors conclude "Overall, these results lead us to suggest that results from TRC experiments should be supplemented with spectroscopic experiments on proteins in solutions and crystalline state...." Line 297-299. As I mentioned in my first review, this is not a new idea and the authors are not the first to do so. The examples especially on fluorescent proteins should be cited, this would also be a good place for the bacteriorhodopsin reference (Line 53-55).

In conclusion, the manuscript is much improved but still too superficial. It is a pity. The authors should try harder to focus on what is new.

Minor points:

Line 53-55: The sentence about bacteriorhodopsin is out of place here and should be moved elsewhere.

Line 93: IRF? Not introduced

Figure 1 A,B. It would be very helpful if the spectra were plotted using the same scale on the x-axis (wavelength) to facilitate direct comparison of the spectra. What is the reason that the isosbestic point differs so much?

Line 129: to be highest for? [instead of from]

Reviewer #2 (Remarks to the Author):

The authors rewrote the manuscript, and the paper has become readable. This change made their results and arguments clearer but also disclosed the weakness of this paper. I think that this paper is publishable in a specialized journal but does not merit publication in Nat. Comm. I mention the weakness of this paper in the following.

1. This paper presented elaborate comparison between PYP dynamics in solution (PYPs) and crystal

(PYPc) in a very wide time range from picosecond to millisecond. Time-absorption spectroscopy shows "rough" similarity between the dynamics of PYPs and PYPc, but details are noticeable different, as expected. This a nice demonstration but they failed to clarify the origin of the difference. They showed one example about a possible structural difference of the first ground state I0 state with time-resolved IR spectrum of the ¹³C analogue. Nevertheless, I0 itself is not the main focus of this paper, even if it is an element, so that it cannot make this paper worth publishing in Nat. Comm. I completely agree to the message of this paper, "even ultrafast TRC results cannot uncritically extrapolated to in vivo function", but I believe that this has already become a consensus in the community.

2. Although I have already mentioned it in my comments for the original manuscript, I repeat saying that it is necessity to show the original time-resolved infrared absorption spectra in this paper. I know well the usefulness of SADS but the data before any analysis are the most important. Actually, the authors show both original time-resolved spectra and SADS for time-resolved UV-visible absorption. Why don't they do the same for time-resolved infrared data? Seeing the temporal traces given in SI, I start suspecting that the authors are not willing to show them because of the low S/N. If this is the case, it is not acceptable.

3. In addition to the above-mentioned problems, the scholarly presentation of this paper has many problems. The authors might say that it is nothing to do with scientific quality but the scholarly presentation is also crucial if they wish to publish their work in a premier journal. I only list a few examples in the following: the figure number is different from the order of the appearance (Figures 2 and 3); Combination of the four figures in Figure 2 does not well match the descriptions in the main text; Both of original time-resolved spectra and SADS are given for transient UV-visible absorption but only SADS is given for transient infrared absorption; a fatal typo about figure number, i.e., Figure 2B,C reads Figure2C,D in Line141; inconsistent number about the vibrational frequency between the main text and figure, i.e., 1685 cm⁻¹ (Line104) vs. 1684 cm⁻¹ (Figure 3).

Reviewer #3 (Remarks to the Author):

I have read the response to reviewers and the revised paper. From my (reviewer 3) point of view the main matters have been addressed. With reviewer 2 I am pleased to see that actual data has been included, and forms the basis for the initial mechanistic discussion. This adds certainty to the main conclusion that TRC and TR solutions experiments can provide different dynamics. As TRC becomes more prevalent realizing this will become more pressing.

I see from other comments that there are shades of opinion on assignment of structures to intermediates in PYP. I am not competent to assess such specialist points for PYP.

In my view the paper can be published, and is an important contribution to field of interpreting TRC data.

Thank you for the opportunity to reply to the comments of the reviewers. We thank the reviewers whose comments have helped us to further improve our manuscript.

In response to the comments by the reviewers we submit a revised version of the manuscript in which we have:

- a) Included the raw absorption infrared absorption difference spectra, as figure 2.
- b) Moved the full target analysis schemes (fig S7) from the SI to the main article, as figure 4.

In the reply to the reviewers we provide a point-by-point response explaining how we have addressed each of the reviewers' comments in blue and repeat the reviewers' remarks in black.

Reviewer #1 (Remarks to the Author):

The revised manuscript by Konold et al has undergone substantial changes, is much improved and most of my concerns have been addressed. However, the authors should focus more on what is new. It is well known, or at least published, that the photo cycle differs in solution and crystals. In the latter there are additional states such as pR1 (see e.g. Jung et al), and the slower time scales deviate quite a bit due to crystal packing constraints. In the literature there have been different results of photocycle kinetics published in studies focusing either on solution or on crystals. This has led to quite a controversy and discussion on whether these diverging results are due to the different techniques used (sensitive to different events in the photocycle), to the modeling/interpretation, to sample preparation, or to the mesoscopic state of the protein i.e. in crystalline or solution state. Here, for the first time, we provide a one-to-one comparison between crystallized and solution PYP, and show that the dynamics in each are different. Upon fitting the data, we find that extra photocycle intermediates are formed in the crystal. This resolves much of the discussion in literature which has been going on since the start of TRX experiments. Of course, some people may have hypothesized that indeed it was the crystalline state of the protein that caused these differences all along, but this is the first time it has experimentally been shown to be the case.

I realize that the photocycle of PYP is complex and that the different nomenclatures do not help. However, given the focus of the manuscript, differences on the ultrafast time scale in solution/crystal, I feel that it would be more appropriate to show Supplementary Fig. S6 in the main text than the strongly simplified Fig. 2. Even more so as the intricacies of Fig. S6 are described in the main text. It seems to me that the schemes shown in Fig. S6 for solution and crystal are rather similar? If not this needs to be stressed further and more explicitly than currently done. It is a pity that the authors have not made more use of the large amount of space available in Nature Comm.

We agree with this comment and we have now included the full photocycle schemes in the main text, as figure 4. On p. 8 we say: "different parameters were required to achieve an adequate fit for each PYP_S and PYP_C, and, notably, the crystalline kinetic model included an additional intermediate state on the nanosecond time scale." This summarizes the main differences between the photocycle schemes. On p. 11 and 12 we dedicate a paragraph to the differences between the two models in more detail: "In accordance with the observed differences in the time traces, fitting of the data against the target model yields different rates for PYP_S and PYP_C, and an extra intermediate was required for PYP_C. The initial decay of the ES in PYP_C is fastest, 0.35 picoseconds, but results in only 10% pR0. A second decay phase in the ES leads to further pR0 formation in 2.6 picoseconds, resulting in an overall quantum yield of 23% (Figure 4). For clarity, we also present a simplified version of the photocycle, in Figure 6. There, we indicate the excited state process of PYP_C with a weighted average of 1 picosecond. In PYP_S, 29% of I0 is formed in 0.6 picoseconds and another 2% in 2 picoseconds. We summarize this as, formation of the first cis isomer of pCa, with red-shifted product absorption, is slower in PYP_C than in PYP_S (1 versus 0.6 picoseconds) and is formed with a lower quantum yield (0.23 vs 0.31). In the formation and dynamics of the initial red-shifted cis-isomer photoproducts in PYP_C an additional intermediate is present: pR0, pR1, pR2 vs I0 and I1 in PYP_S. pR0 has a shorter lifetime than I0 (0.3 versus 1 nanosecond) and pR1 decays into pR2 in 18 nanoseconds. The formation of pB-like states occurs on similar timescales in PYP_S and PYP_C, but with a lower quantum yield, as only half of the pR2 states form pBC, resulting in a pB yield of 0.3 versus 0.11 in PYP_S and PYP_C respectively. Recovery of the ground state takes 9 milliseconds in PYP_C and occurs with biphasic time constants of 1.3 and 320 milliseconds in PYP_S." We trust that this is sufficient to point out the differences between the two models.

Along these lines, I find the manuscript rather difficult to read. It is not always clear which sample is being described (crystalline/solution). I understand that the authors want to directly compare features between the two samples instead of describing one sample first, then the other, then the differences but this comes at a price. Not sure it is worth it.

In the current version of the manuscript we have revised the figures. We understand that several permutations of presenting the results are possible. We feel that as the article is focused on the comparison between PYP_S and PYP_C, it is best to show results of each side by side for each experiment.

I think a stringent comparison of the ultrafast photocycle in solution and crystals would be interesting.

This is indeed what is being reported in this manuscript on p. 11 and 12 of the manuscript, and further discussed on p. 18-19.

It is important that comparable parameters are used for the experiments. It is therefore very good that the authors also pointed out the reference describing the lack of influence of increasing the PEG concentration on the photocycle kinetics. This limits the reasons for the apparent discrepancy.

There is one further difference that the authors did not mention: The orientation of protein embedded chromophores in solution and crystals differ, they are randomly distributed with respect to the photoexcitation polarization plane in solution whereas photo selection rules likely apply in oriented crystals (see e.g. page 19260 PYP photoactivation; Schotte et al, PNAS). The crystalline pancake approach, making a sample for spectroscopy by squishing a very large crystal between glass plates separated by thin spacers may not be good enough to randomize the orientations of the microcrystalline fragments. This could be avoided by stirring up the squished crystal and letting the crystallites settle. I could understand if the authors did not want to repeat some of the measurements using such a sample, but they should address this problem in the manuscript thoroughly. Or add a disclaimer that this could influence the kinetics in the crystalline state. I take it they use linearly polarized light? (not stated). SFX experiments suffer less from this issue because they use randomly oriented crystals for data collection and most often circularly polarized light.

The reviewer is right that it is useful to comment on this issue. In the experiments we have used a configuration where pump and probe polarization are under the magic angle. Prompted by this remark of the reviewer we have added the following sentence to the M&M: "The polarization of pump and probe beams was under the magic angle (54.7°). In combination with the crushing procedure of the crystals, this minimizes possible photo-selection effects in the crystals."

The authors are likely aware that there are a number of controversies associated with the time-resolved crystallographic data. By no means do Schotte et al and Jung et al agree, see controversy published in Nature Chemistry 2014, this concerns the conformation of the chromophore and the lack of observation of pR1 by Schotte et al. In particular the ultrafast TR-SRX experiments on PYP performed by Schmidt and coworkers used much higher power densities for photoexcitation than considered safe by Hutchinson et al., these experiments were all performed using multiphoton excitation.

In our opinion the reviewer is too strong in stating that the Schotte and Jung studies show controversies. The studies very much agree on the data; their main difference is in the bond-order of the double bond that can isomerize in the chromophore. To clarify this, on p 12 we have inserted the following sentence: " (Note that these latter three studies mutually agree on the data, but differ in the bond-order of the double bond that can isomerize in the chromophore, due the use of DFT optimized structure vs non-DFT optimized structures.46)"

Photoexcitation was mostly performed at 450 nm, the PYP crystals were grown from salt and not PEG. I am therefore surprised by the statement that "the PYPc results closely follow those by Schotte, Jung and Tenboer [misspelled]. This included" and the 22 % overall pR quantum yield" ® Line 175-178. The quantum yield in PYP is wavelength dependent, why would it be the same?

Tenboer has been corrected in Ten Boer.

The paper of L. Tyler Mix, Elizabeth C. Carroll, Dmitry Morozov, Jie Pan, Wendy Ryan Gordon, Andrew Philip, Jack Fuzell, Masato Kumauchi, Ivo van Stokkum, Gerrit Groenhof, Wouter D. Hoff and Delmar S. Larsen Biochemistry, 57(11), 1733-1747 (2018) studied PYP dynamics as a function of wavelength and resolved conformational heterogeneity. They show that there is a slight wavelength-dependence of the quantum yield of **fluorescence** of PYP, but virtually no wavelength dependence of the quantum yield of photochemistry of PYP. The finding that the quantum yields for PYPS and PYPC differ, but for PYPC is similar to that found for PYPC in Ten Boer we consider worth noting.

Line 244: "Changes in the region of the " "this could be checked in the various crystal structures (e.g. see Table 1, Pande et al), with the caveats mentioned above

We thank the reviewer for this suggesting and added: " ... is relaxed throughout the photocycle of PYPC, in agreement with the reported lengthening from 2.50 to 2.94 Å in the TRX study of Pande et al 5, but in contrast to the observed strengthening in the PYPS I0 and I1 states 35,38."

Line 250: "Schotte et al reported that in pB, Arg52 switches to an open position.."± This is correct. But Arg52 moves because it would otherwise directly overlap with the oxygen of the cofactor in the new pCa conformation in pB. The pCa oxygen is 2.9 Å from a water molecule (2115) and 2.9 Å from NE of Arg52. Glu46 forms a hydrogen bond to a water molecule(2056) located at the position of the oxygen atom of dark state pCA (3.2 Å). The water also forms a hydrogen bond to Tyr42. (Glu-O-pCa: dark2.5 Å, pR0: 2.7 Å, pR2: 2.6 Å); structures from Schotte.

We thank the reviewer for this suggestion, but we have used the discussion as can be found in the article by Schotte et al, rather than embark on our own interpretation of the Schotte data. We feel that this is the correct way to approach this issue and did not change the text.

Line 259: " We conclude that our transient measurements on PYPC are remarkably consistent with the photocycle elements as observed in the TRC experiment"±. First, see caveats mentioned above. Second, Pande et al observes full isomerization within 0.6 ps, no further changes take place on the ps time scale. This contradicts

the findings described Line 162-165 and 268-270. As mentioned above, there may be many reasons for this discrepancy. But lumping all time-resolved crystallographic experiments together and saying that they agree with the findings of the authors is not adequate for the purpose of this manuscript.

We agree with the reviewer that this statement is too broad. Earlier in the manuscript, we state that our results agree with those of Schotte², Jung³ and Ten Boer et. al¹¹, and we have placed that more accurate statement now also in line 259. The Pande study only measured out to 3 ps therefore was unable to observe the structural relaxation that occurs later in the photocycle, as observed by Schotte and Jung and in combination with the degree of noise in TRX their statement that full isomerization occurred in 0.6 ps in the crystal may have been too strong.

The authors attribute dehydration as the reason for the discrepancies between PYP photocycle kinetics in solution and the crystalline state. Increasing PEG concentrations is used frequently by crystallographers to dehydrate crystals to improve packing. Doesn't the lack of influence of increasing the PEG concentration on the dynamics argue against this? Yeremenko only studied micro to millisecond spectroscopy. In this time range the photocycle differs a lot between solution and crystal anyway. It does not necessarily show that PEG does not influence the kinetics on the fast time scale. This caveat needs to be mentioned. A nice recent example is given by Woodhouse Nature Comm 2020, showing that the presence of ammonium sulfate (used for crystallization) can change the kinetics in a fluorescent protein. Jasper van Thor thinks it is the viscosity of the protein that influences, or at least can influence the ultrafast dynamics upon photoexcitation. He mentions this in one of his publications.

We have rephrased the one but last paragraph in which this is discussed. We now mention the different options that may lie at the molecular basis for the different dynamics in crystalline and solution PYP, but end with: "However, the molecular basis of the different ultrafast parts of the photocycle of PYP and PYPS, whether this is dehydration, altered viscosity or confinement by the crystal lattice remains to be decided and it may well be a mixture of these factors." We did not find Van Thor's publication that the reviewer refers to.

The authors conclude "Overall, these results lead us to suggest that results from TRC experiments should be supplemented with spectroscopic experiments on proteins in solutions and crystalline state". Line 297-299. As I mentioned in my first review, this is not a new idea and the authors are not the first to do so. The examples especially on fluorescent proteins should be cited, this would also be a good place for the bacteriorhodopsin reference (Line 53-55).

The novelty of our work is that we actually side by side *measured* the full photocycle of a protein in crystalline form and in solution, and demonstrate differences on an ultrafast time scale. By doing so we recover dynamics and intermediates earlier resolved in TRX experiments and therefore unequivocally show that it is indeed the protein itself that shows a different photocycle in TRX, and that it is not the result of the use of different measurement techniques but that TRX and spectroscopy are sensitive to the same events in the photocycle, and it is the mesoscopic state of the protein that alters the dynamics.

The bacteriorhodopsin study is now in the last paragraph of the discussion where we inserted a short discussion of power dependent effects on the photocycle..

In conclusion, the manuscript is much improved but still too superficial. It is a pity. The authors should try harder to focus on what is new.

We find this comment hard to understand. This is new. An important reason for us to perform this study, was that a) There is an immense amount of ultrafast spectroscopic data on photoactive proteins, including PYP, b) With the advent of TRC, of many of these proteins also ultrafast, short-lived structures will be resolved. As for those few proteins for which those datasets can be compared, such as PYP, it appears they cannot be reconciled. It is therefore of the utmost importance to experimentally prove that they can be reconciled IF you measure on the same state of the protein, rather than thinking or hypothesizing this is probably the case. Overall, our message is that transient crystallographic and spectroscopic techniques are highly complementary and most effective when applied in a symbiotic fashion in the context of resolving of protein dynamics with varying sample treatment, if you want to relate the dynamics to those in vivo.

Minor points:

Line 53-55: The sentence about bacteriorhodopsin is out of place here and should be moved elsewhere. This has been moved to the last part of the discussion.

Line 93: IRF? Not introduced. Corrected

Figure 1 A,B. It would be very helpful if the spectra were plotted using the same scale on the x-axis (wavelength) to facilitate direct comparison of the spectra. What is the reason that the isosbestic point differs so much?

The spectra have now been plotted on the same scale. On p. 4 we discuss distortions in the crystalline spectra: "... due to the higher absorption in the crystals, diminishing the bleach signals at 440 nm and to increased scatter, which leads to a certain degree of distortion of the absorption difference spectra."

Line 129: to be highest for? [instead of from] Corrected.

Reviewer #2 (Remarks to the Author):

The authors rewrote the manuscript, and the paper has become readable. This change made their results and arguments clearer but also disclosed the weakness of this paper. I think that this paper is publishable in a specialized journal but does not merit publication in Nat. Comm. I mention the weakness of this paper in the following.

1.

This paper presented elaborate comparison between PYP dynamics in solution (PYPs) and crystal (PYPc) in a very wide time range from picosecond to millisecond. Time-absorption spectroscopy shows "rough"± similarity between the dynamics of PYPs and PYPc, but details are noticeable different, as expected. This a nice demonstration but they failed to clarify the origin of the difference. They showed one example about a possible structural difference of the first ground state I0 state with time-resolved IR spectrum of the 13C analogue. Nevertheless, I0 itself is not the main focus of this paper, even if it is an element, so that it cannot make this paper worth publishing in Nat. Comm. I completely agree to the message of this paper, "even ultrafast TRC results cannot uncritically extrapolated to in vivo function"±, but I believe that this has already become a consensus in the community.

In this manuscript we show that the photocycle of PYP in solution is different than in crystallized state, over the full photocycle from 100 femtoseconds over 12 decades in time. This is the first time this has experimentally been shown to be the case in a single experiment, and that differences between solution and crystal have been shown on the ultrafast timescale. The relevance of this we pointed out in our reply to reviewer 1: . An important reason for us to perform this study, was that a) There is an immense amount of ultrafast spectroscopic data on photoactive proteins, including PYP, b) With the advent of TRC, of many of these proteins also ultrafast, short-lived structures will be resolved. As for those few proteins for which those datasets can be compared, such as PYP, it appears they cannot be reconciled. It is therefore of the utmost importance to experimentally prove that they can be reconciled IF you measure on the same state of the protein, rather than thinking or hypothesizing this is probably the case. Overall, our message is that transient crystallographic and spectroscopic techniques are highly complementary and most effective when applied in a symbiotic fashion in the context of resolving of protein dynamics with varying sample treatment, if you want to relate the dynamics to those in vivo.

2.

Although I have already mentioned it in my comments for the original manuscript, I repeat saying that it is necessary to show the original time-resolved infrared absorption spectra in this paper. I know well the usefulness of SADS but the data before any analysis are the most important. Actually, the authors show both original time-resolved spectra and SADS for time-resolved UV-visible absorption. Why don't they do the same for time-resolved infrared data? Seeing the temporal traces given in SI, I start suspecting that the authors are not willing to show them because of the low S/N. If this is the case, it is not acceptable.

On request of the reviewer, we have now also included the time-resolved spectra of the infrared data. As the reviewer will see the S/N is actually very good. Most of the noise is so-called baseline noise (as we measure full spectra in a single laser shot), which affects the time traces, which we showed all along, much more than the whole spectra.

3.

In addition to the above-mentioned problems, the scholarly presentation of this paper has many problems. The authors might say that it is nothing to do with scientific quality but the scholarly presentation is also crucial if they wish to publish their work in a premier journal. I only list a few examples in the following: the figure number is different from the order of the appearance (Figures 2 and 3); Combination of the four figures in Figure 2 does not well match the descriptions in the main text; Both of original time-resolved spectra and SADS are given for transient UV-visible absorption but only SADS is given for transient infrared absorption; a fatal typo about figure number, i.e., Figure 2B,C reads Figure2C,D in Line141; inconsistent number about the vibrational frequency between the main text and figure, i.e., 1685 cm⁻¹ (Line104) vs. 1684 cm⁻¹ (Figure 3).

We thank the reviewer for pointing out these errors and have remade the figures and checked their referencing in the text.

Reviewer #3 (Remarks to the Author):

I have read the response to reviewers and the revised paper. From my (reviewer 3) point of view the main matters have been addressed. With reviewer 2 I am pleased to see that actual data has been included, and forms the basis for the initial mechanistic discussion. This adds certainty to the main conclusion that TRC and TR solutions experiments can provide different dynamics. As TRC becomes more prevalent realizing this will become more pressing.

I see from other comments that there are shades of opinion on assignment of structures to intermediates in PYP. I am not competent to assess such specialist points for PYP.

In my view the paper can be published, and is an important contribution to field of interpreting TRC data.

We thank the reviewer for this favorable assessment.